# The yeast genome is globally accessible in living cells

Hemant K. Prajapati [1,3], Peter R. Eriksson[1,3], Paul A. Elizalde[1,2], Christopher T. Coey[1], Zhuwei Xu[1] & David J. Clark [1] ✉

Eukaryotic genomes are packaged into chromatin, which is composed of condensed filaments of regularly spaced nucleosomes, resembling beads on a string. The nucleosome contains ~147 bp of DNA wrapped almost twice around a central core histone octamer. The packaging of DNA into chromatin represents a challenge to transcription factors and other proteins requiring access to their binding sites. Consequently, control of DNA accessibility is thought to play a key role in gene regulation. Here we measure DNA accessibility genome wide in living budding yeast cells by inducible expression of DNA methyltransferases. We find that the genome is globally accessible in living cells, unlike in isolated nuclei, where DNA accessibility is severely restricted. Gene bodies are methylated at only slightly slower rates than promoters, indicating that yeast chromatin is highly dynamic in vivo. In contrast, silenced loci and centromeres are strongly protected. Global shifts in nucleosome positions occur in cells as they are depleted of ATP-dependent chromatin remodelers, suggesting that nucleosome dynamics result from competition among these enzymes. We conclude that chromatin is in a state of continuous flux in living cells, but static in nuclei, suggesting that DNA packaging in yeast is not generally repressive.

The packaging of chromosomal DNA in eukaryotic nuclei occurs at several structural levels. The most basic structure resembles beads on a string, in which the beads are nucleosome cores and the string is the connecting linker DNA[1]. The nucleosome is composed of 145–147 bp of DNA tightly wrapped nearly twice in a negative superhelix around a central protein core containing two molecules each of the four core histones (H2A, H2B, H3 and H4) (ref. 2). The next level of structure is the spontaneous coiling of the nucleosomal filament into a heterogeneous higher-order structure ~30 nm wide, facilitated by linker histones[3]. The nucleosomal filament is further organized into loop domains by other chromatin proteins, such as CTCF and condensin[4]. Finally, inactive chromatin (heterochromatin) is generally much more condensed than active chromatin (euchromatin) and is associated with additional compacting proteins, such as HP1 (ref. 5).

The packaging of the genome presents a formidable obstacle to DNA-binding proteins searching for their specific binding sites[6]. Indeed, many in vitro studies have confirmed the generally repressive nature of chromatin[7]. Most sequence-specific transcription factors bind very poorly, if at all, to a cognate site within a nucleosome, unless the site is located just inside the nucleosome, where the histone–DNA contacts are weakest[8]. The pioneer factors are exceptions, having evolved to bind nucleosomal sites with high affinity[9]. The general inhibition of transcription factor binding by nucleosomes in vitro led to the discovery of ATP-dependent chromatin remodeling enzymes, which facilitate access to the DNA[10,11]. Their activities include sliding nucleosomes along DNA, transferring the histone octamer from one DNA molecule to another or inducing conformational changes in nucleosomes[12,13]. Examples include the yeast SWI/SNF, RSC, ISW1

[1]Division of Developmental Biology, Eunice Kennedy-Shriver National Institute of Child Health and Human Development, National Institutes of Health, Bethesda, MD, USA. [2]Present address: NIH–JHU Graduate Partnership Program, Johns Hopkins University, Baltimore, MD, USA. [3]These authors contributed equally: Hemant K. Prajapati, Peter R. Eriksson. ✉e-mail: clarkda@mail.nih.gov

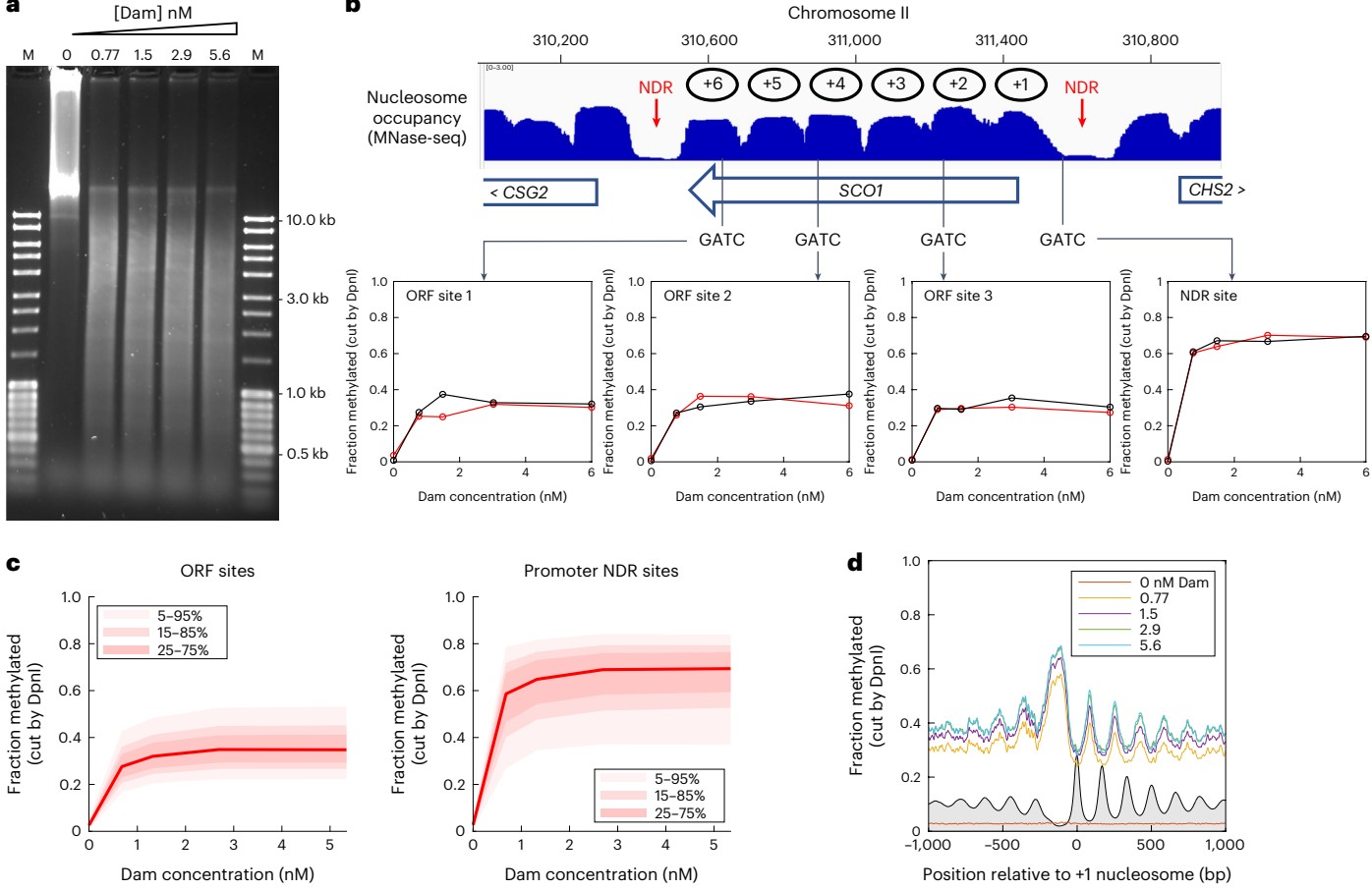

**Fig. 1 | Access to the yeast genome is limited in nuclei. a**, Genomic DNA is only partially methylated by Dam in nuclei. Nuclei were treated with increasing concentrations of Dam (0, 0.77, 1.5, 2.9, 5.6nM); purified genomic DNA was digested with DpnI. Electrophoretic analysis in a 1% agarose gel stained with SYBR Gold. M, DNA size marker. **b**, Limited methylation of specific GATC sites in the *SCO1* gene. Nucleosome occupancy was determined by MNase-seq data[26]. Methylation as a function of Dam concentration for each of the four sites in *SCO1* (red, replicate 1; black, replicate 2). **c**, Global analysis of GATC sites in ORFs and promoter NDRs (defined using MNase-seq data). The red line and shading depicts median GATC site with data range indicated. **d**, Nucleosome phasing detected by Dam methylation. Data obtained at different Dam concentrations for GATC sites in all ~5,770 yeast genes are plotted relative to the +1 nucleosome dyad (smoothed with a 21 bp window). Gray profile, nucleosome dyad distribution (MNase-seq data[26] normalized to 0.1).

and CHD1 multisubunit complexes and their metazoan counterparts, including human BAF and PBAF, SNF2H/L and complexes containing Chd1 homologs. These enzymes play a critical role in gene regulation; they act at promoters and other regulatory elements to facilitate the creation of nucleosome-depleted regions (NDRs)[14]. Mutations in genes encoding chromatin remodeler subunits have been associated with many different human diseases, including many cancers[15–17] and autism spectrum disorder[18].

Genome-wide studies of nuclei from various organisms have revealed a link between the nucleosome occupancy of a promoter and gene activity: active promoters tend to be nucleosome depleted, whereas inactive promoters are occupied by nucleosomes (reviewed in ref. 19, but see ref. 20). This observation supports a widely accepted model positing that chromatin blocks access to DNA, particularly regulatory elements such as promoters and enhancers, preventing transcription factors from binding to their cognate sites and activating transcription[6,19]. However, nuclei are necessarily derived from disrupted cells, and the loss of critical cofactors, such as ATP, might be expected to 'freeze' the chromatin structure at the moment of cell breakage. So, do experiments with nuclei report the true nature of chromatin in vivo? In this Article, we address this question by adapting our quantitative DNA accessibility sequencing assay (qDA-seq) method[20], developed for quantitative genome-wide measurement

of DNA accessibility in nuclei, for use in living budding yeast cells. We expressed the sequence-specific *Escherichia coli* dam DNA adenine methyltransferase (Dam) from an inducible promoter. Dam methylates the adenine in the sequence GATC, which, if methylated on both strands, can be cut by the restriction enzyme DpnI.

## Limited access to the genome in nuclei

Previously, we used qDA-seq to measure DNA accessibility in haploid yeast nuclei using the restriction enzyme AluI. If its cognate site (AGCT) is inside a nucleosome, AluI is blocked, but if it is in linker DNA, it is cleaved, creating a double-strand break. We measured the extent of digestion as a function of AluI concentration at all ~40,000 AGCT sites in the yeast genome. We found that virtually all AluI sites reach a limit digest, indicating that they are accessible in some nuclei and inaccessible in the rest. AluI sites located in promoters are accessible in about half of the nuclei, whereas AluI sites in gene bodies are accessible in about a quarter of nuclei (that is, inaccessible in ~75% of nuclei). This observation can be explained by imperfect nucleosome positioning, such that each site is in a linker in some nuclei and nucleosomal in the others.

We sought to determine whether DNA accessibility in living cells is similar to that in nuclei. Since expression of AluI in live cells is likely to be lethal due to double-strand breaks, we used Dam methylase as

a probe instead. First, we confirmed that Dam gives similar results to AluI for nuclei (Fig. 1). Purified wild-type nuclei were incubated with increasing amounts of Dam for 20 min at 25 °C. Genomic DNA was purified, digested with excess DpnI and analyzed in an agarose gel (Fig. 1a). Partial digestion was observed, reaching a limit even at the highest Dam concentration (cf. the complete MboI digest of unmethylated genomic DNA by MboI, which recognizes unmethylated GATC; Fig. 2c). This result is similar to our AluI data, indicating that much of the genome is inaccessible to Dam in nuclei.

DpnI-digested DNA was fragmented by sonication and used to prepare libraries for paired-end sequencing. We calculated the fraction cut ('fcut') by DpnI (that is, methylated by Dam) at each Dam concentration at each of the 35,830 GATC sites in the yeast genome by counting the number of ends coinciding with a GATC site and dividing by the fragment coverage for that site (each side of the cut was counted separately; Methods). A plot of fcut as a function of Dam concentration is generated for all of the sites. The four GATC sites located in the *SCO1* gene were selected for illustration (Fig. 1b). The site located in the *SCO1* promoter NDR reaches a limit of ~70% methylation; higher Dam concentrations do not result in more methylation. Thus, this site is accessible to Dam in ~70% of nuclei, but protected in the remaining 30%. Similar plots for the three sites located in the *SCO1* open reading frame (ORF) exhibit lower limit values ranging from ~30% to 40%, consistent with the presence of a blocking nucleosome at each site in 60–70% of nuclei.

To display the data for sites in different genomic regions at the global level, we superimposed the plots for all GATC sites located in each region and calculated the median methylation (Fig. 1c and Extended Data Fig. 1a,b). At least 90% of ORF sites are only partially methylated, reaching a limit of ~30% methylation, indicating that the median ORF site is accessible in 30% of nuclei and inaccessible in the remaining ~70% (consistent with data for nuclei treated with other DNA methylases[21]). A limit digest is also reached at promoter NDRs, but at a higher value of ~70%, indicating that the median promoter is accessible in 70% of nuclei and inaccessible in the rest. We argued previously that the observed protection in ORFs is primarily due to nucleosomes, whereas protection in NDRs is due to stable transcription complexes[22,23].

We investigated the variation in accessibility along the average gene in nuclei by plotting the mean value of fcut as a function of distance from the dyad of the first (+1) nucleosome for all genes (Fig. 1d and Extended Data Fig. 1c). In the absence of Dam, there is a very low level of apparent methylation (~3%), representing the probability of random DNA fragmentation by sonication at each nucleotide. In the presence of Dam, the promoter NDR is associated with a peak reaching a limit at ~70%. The observed methylation pattern is exactly out of phase with the nucleosome phasing pattern obtained by micrococcal nuclease digestion with deep sequencing (MNase-seq) for nuclei from the same strain, indicating that methylation is high in linkers and low within nucleosomes. These data show that nucleosome positioning is far from perfect, such that the MNase peaks represent the most commonly occupied positions in the cell population. We conclude that the accessibility of genomic DNA is strictly limited in nuclei.

## Global access to the genome in vivo

To express Dam in vivo, we used a modified *CUP1* promoter containing a single high-affinity Gcn4 transcription factor binding site[24]. In the absence of isoleucine and valine, the addition of sulfometuron methyl (SM) stimulates translation of the *GCN4* messenger RNA, resulting in Gcn4-induced transcription[25]. To reduce background methylation before induction, we fused Dam to an auxin-dependent degron, and included *TIR1* (encoding the ubiquitin-specific ligase required for auxin dependence) in the integration cassette (Fig. 2a). SM induction produces a Dam–degron-3HA fusion protein, which is degraded if auxin is present.

The experiment involves growth in auxin to maintain low background methylation, followed by auxin removal and a time course of SM induction. Robust Dam induction was observed by immunoblotting

(Fig. 2b). Analysis of DpnI-digested genomic DNA from each time point in an agarose gel showed that digestion is almost complete after 240 min of SM treatment, indicating that virtually the entire genome had been methylated and was therefore accessible to Dam in living cells (Fig. 2c), unlike in nuclei (Fig. 1a). This conclusion is supported by genomic analysis: the four *SCO1* sites are almost completely methylated (Fig. 2d), as are the median ORF and promoter NDR sites (Fig. 2e). Even though auxin is present before SM addition, there is some background methylation, which is higher for sites in promoter NDRs than for those in ORFs. Analysis of nucleosome phasing in living cells shows the same pattern that is observed in nuclei, but the average methylation level increases with time until all genic sites are almost completely methylated (Fig. 2f and Extended Data Fig. 2c). This observation indicates that the average nucleosomal organization of genes is preserved in cells over time, but Dam can still access nucleosomal DNA. We conclude that, unlike in nuclei, genome accessibility is not limited in living cells.

To determine whether transcriptional activity promotes methylation, we divided all genes into deciles according to their RNA polymerase II levels, as measured for this strain by chromatin immunoprecipitation sequencing (ChIP-seq) for the Rpb3 subunit[26], and plotted their median methylation rates (Fig. 2g and Extended Data Fig. 2d). We observed a very small but reproducible trend of increasing median methylation with increasing transcriptional activity, such that the most active genes (decile 1) are methylated slightly faster than the least active genes (decile 10). We also examined Gcn4-activated genes[27], which are induced by SM at the same time as Dam. The ORFs of these genes are methylated marginally faster than other genes (Extended Data Fig. 2a). A small subset of Gcn4-activated ORFs with highly disrupted chromatin (including some H2A–H2B dimer loss) correlated with high RNA polymerase II levels[28,29] is methylated faster than the full set of Gcn4 target genes, indicating that very highly transcribed genes are more accessible.

SM-treated cells grow very slowly, but they still undergo cell division during the time course. We addressed the potential role of DNA replication in creating genome accessibility by repeating the experiment with cells arrested in G1 using α-factor. Cells growing in auxin were treated with α-factor for 3 h, then auxin was removed and SM was added for a 4 h time course in the continued presence of α-factor (Extended Data Fig. 3a). Genome accessibility in arrested cells (Extended Data Fig. 3b–f) is similar to that in growing cells (Fig. 2). We conclude that global accessibility is not due to replication.

## Cell-to-cell variation in Dam expression

To gain insight into the methylation time course at the single-cell level, we fused green fluorescent protein (GFP) to Dam and repeated the experiment (Extended Data Fig. 4a). The rate of ORF methylation at the median GATC site by Dam–degron–GFP-3HA is similar to that for Dam–degron-3HA, although immunoblotting showed that the fusion protein is cleaved between the Dam and GFP domains (Extended Data Fig. 4b–d). After SM addition, individual cells become GFP positive at different times (Extended Data Fig. 4e). The fraction of GFP-negative cells decreases exponentially, such that half of the cells produce detectable GFP only after ~80 min of SM treatment (Extended Data Fig. 4f). A plot of the median ORF site methylation against the fraction of GFP-positive cells yields a straight line with $R^2 > 0.97$, showing that median ORF methylation is directly proportional to the fraction of GFP-positive cells (Extended Data Fig. 4g). Similarly, the unmethylated fraction decreases exponentially (Extended Data Fig. 4h). Thus, methylation occurs at different rates in different cells in the population, involving variable delay times before induction, perhaps due to transcriptional bursting of the Dam gene[30], and variable amounts of Dam synthesis, as shown by cell-to-cell variation in brightness (Extended Data Fig. 4e). The amount of Dam produced by the cell population as a whole, measured by the anti-HA blot (Fig. 2b), reflects both the fraction of expressing cells and the amount of Dam per cell, and the median methylation curve represents the sum of all the curves for individual

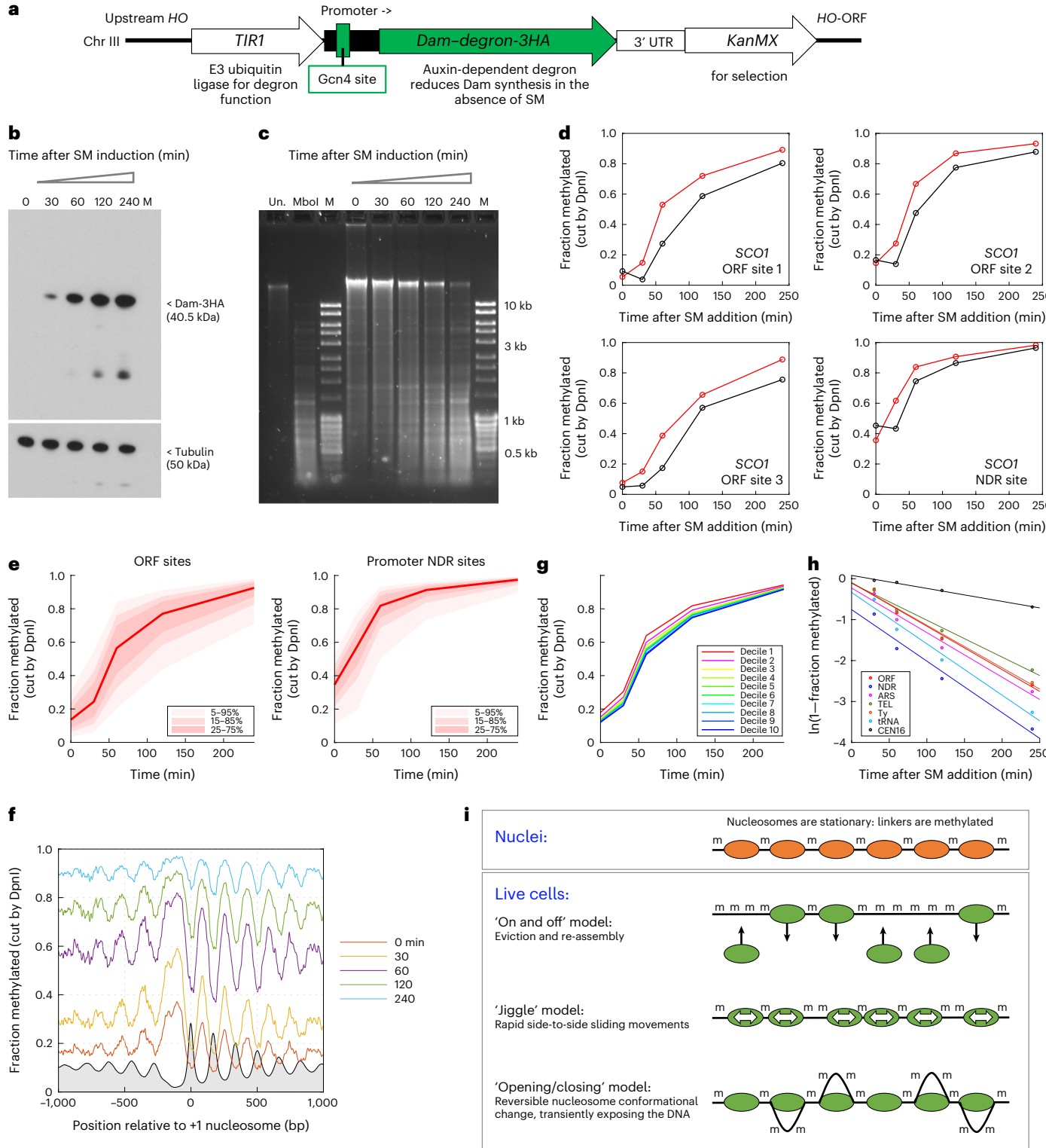

**Fig. 2 | The yeast genome is globally accessible in living cells. a**, Integration construct for SM-inducible Dam expression. **b**, SM induction of Dam–degron-3HA. Anti-HA immunoblot displayed. **c**, Genomic DNA is mostly methylated after Dam induction. Un, undigested genomic DNA; MboI, unmethylated genomic DNA completely digested with MboI (which cuts unmethylated GATC); M, DNA size marker. **d**, Almost complete methylation of specific GATC sites in the *SCO1* gene is shown with methylation time course for each of the four sites in *SCO1* (cf. Fig. 1b): red, replicate 1 and black, replicate 2. **e**, Global analysis of ORF and promoter NDR GATC sites. The red line and shading shows median GATC site with data range indicated. **f**, Nucleosome phasing is detected by Dam methylation: nucleosomal DNA is gradually methylated. Methylation data for GATC sites in all genes at each

time point are plotted relative to the +1 nucleosome dyad (smoothed with a 21 bp window). Gray profile: nucleosome dyad distribution in nuclei (MNase-seq data normalized to 0.1). **g**, Transcriptional activity has a minor effect on methylation rate. The ~5,770 yeast genes were divided into deciles according to their RNA polymerase II levels (using ChIP-seq data for the Rpb3 subunit[29]). Data for the median GATC site in each decile are shown; the most active genes are in Decile 1. **h**, Genomic regions are methylated at quite similar rates, except for the GATC site in *CEN16*. Plot of ln(median unmethylated fraction) for GATC sites in each region against time after SM induction. ARS, autonomous replicating sequences; TEL, telomeric regions; Ty, Ty transposable elements. **i**, Chromatin flux model to explain the difference in genome accessibility in nuclei and living cells.

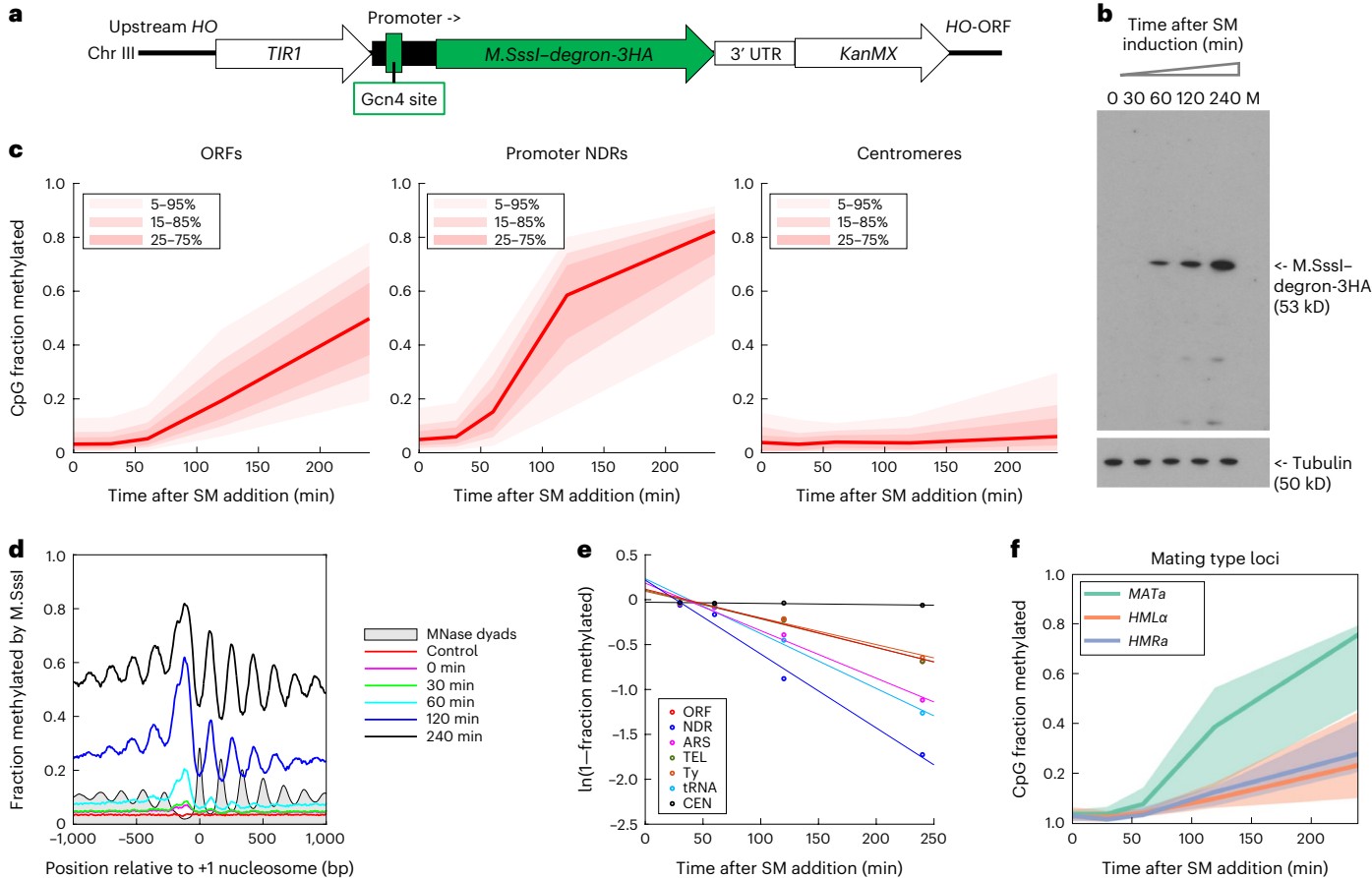

**Fig. 3 | Centromeres and SIR-repressed loci are protected from methylation in living cells. a**, Integration construct for SM-inducible M.SssI expression. **b**, SM induction of M.SssI–degron-3HA is shown with anti-HA immunoblot. **c**, Extensive methylation of CG sites in ORFs and promoters but not centromeres. Methylated CG sites were detected using nanopore long-read sequencing. The red line and shading shows median CG site with data range indicated. **d**, Nucleosome phasing detected by CG methylation. Nucleosomal DNA is extensively methylated. Methylation data obtained for CG sites in all genes at each time point are plotted relative to the +1 nucleosome dyad (smoothed with a 21 bp window). The gray profile shows nucleosome dyad distribution in nuclei (MNase-seq data normalized to 0.1). **e**, Methylation rates for various genomic regions. Plot of ln(median unmethylated fraction) for CG sites in each region against time after SM induction. **f**, Methylation of the silenced *HMLα* and *HMRa* loci is much slower than at the active *MATa* locus. Each line indicates the median CG site with the data range indicated by shading (25–75%).

## Centromeres and silenced loci are protected

We estimated the methylation rates for various genomic regions by plotting the log of the unmethylated fraction against time (Fig. 2h). The slope of the regression line gives an apparent rate constant. The median promoter NDR site is methylated ~1.3 times faster than the median ORF site (Extended Data Fig. 2b,e). Similarly, transfer RNA (tRNA) genes, which are occupied by TFIIIB–TFIIIC transcription complexes rather than nucleosomes[31], are methylated ~1.2 times faster than ORF sites, whereas replication origins (*ARS* elements), Ty transposable elements and telomeric regions are methylated at about the same rate as ORF sites (Extended Data Fig. 2b,e). Thus, promoter NDRs and tRNA genes are methylated only marginally faster than ORF sites. However, if the cell induction rate (Extended Data Fig. 4) is similar to the methylation rate within the cell, a larger difference between NDRs and nucleosomal sites could be obscured.

Although all of the genomic regions discussed above are methylated at quite similar rates, we noted that the single GATC site located just inside the centromeric nucleosome on chromosome 16 (*CEN16*) is methylated much more slowly (Fig. 2h and Extended Data Fig. 2e). In budding yeast, each of the 16 chromosomes has a single centromeric

nucleosome, which contains an H3 variant (Cse4/CenH3), and is precisely positioned on the centromere[32,33]. However, since we had information on only one of the 16 centromeres, the question arose as to whether all centromeres are similarly protected. Noting that all yeast centromeres have at least one 'CG' site, we constructed a yeast strain in which the Dam ORF was replaced with the ORF for M.SssI, a DNA methyltransferase that methylates CG to m⁵CG (Fig. 3a). SM induction of M.SssI is slower than for Dam (Fig. 3b). To detect m⁵C in CG motifs genome wide, we used nanopore long-read sequencing. Median methylation of CG sites in ORFs and promoters is not as high as for Dam (Fig. 2e), but the trend is sharply upward (Fig. 3c).

The different methylation rates exhibited by Dam and M.SssI could be due to differences in enzyme concentration, enzyme turnover number and/or the number of potential methylation sites. In addition, methylation by M.SssI in the absence of SM is much lower. M.SssI methylates promoter NDR sites ~2.9 times faster than ORF sites, whereas the ratio for Dam is ~1.3 (Extended Data Figs. 2e and 5e). To account for this difference, we propose that, at any moment in time, the NDR belonging to each gene is virtually protein free in a fraction of cells and occupied by a nonhistone complex in the remaining cells[22]. If the NDR is protein free, it is methylated rapidly even at the low Dam levels before SM addition (M.SssI levels are too low), whereas, if the NDR is occupied, methylation is slower, occurring only when the occupying complex dissociates.

cells with time. It follows that the methylation rate within the average cell is much faster than the time course for the whole population.

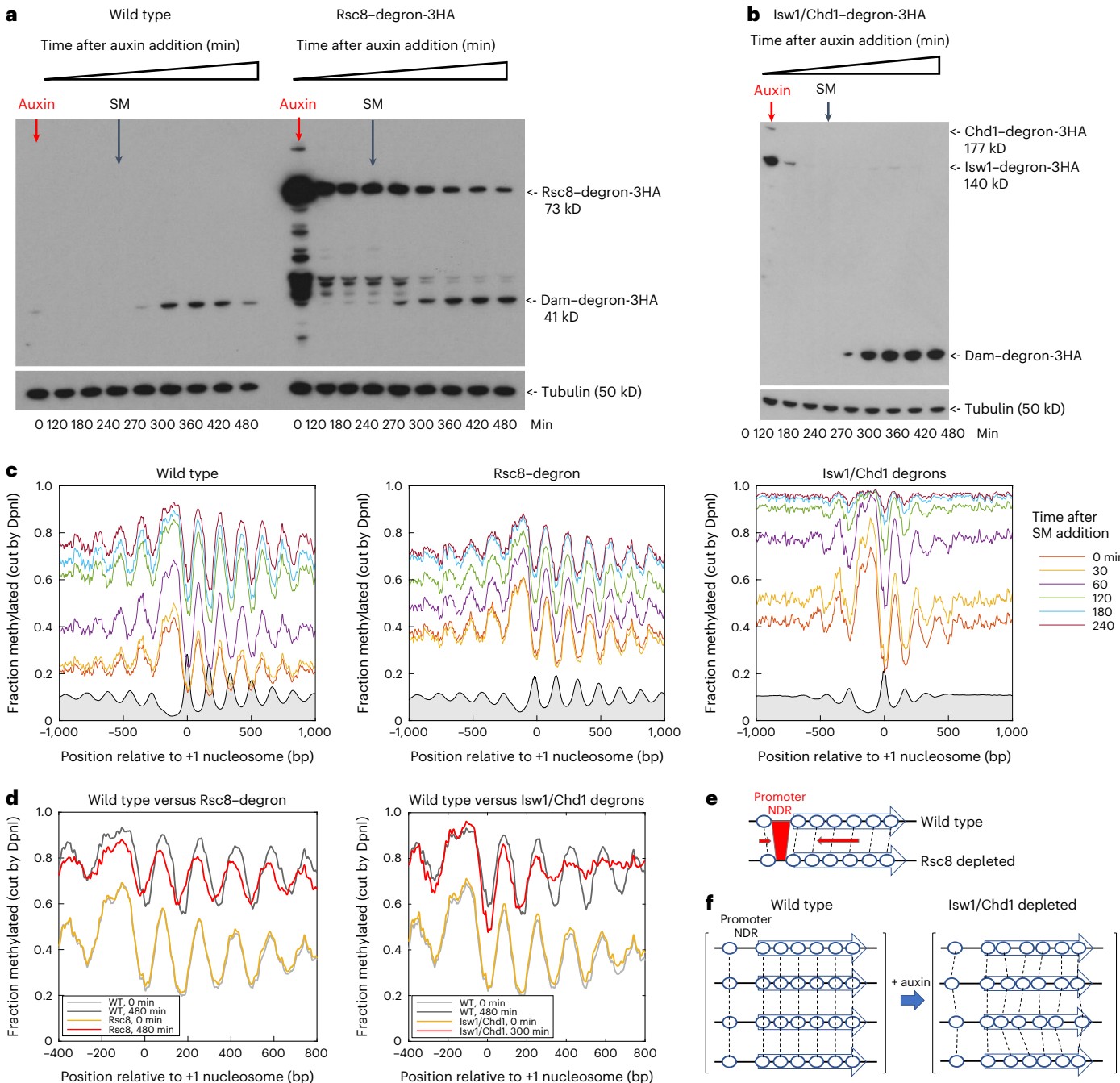

**Fig. 4 | The RSC, ISW1 and CHD1 ATP-dependent chromatin remodelers contribute to global nucleosome mobility in living cells. a**, Anti-HA immunoblot performed to follow Rsc8 depletion and Dam induction. Wild-type and Rsc8–degron cells were treated with auxin for 4 h and then induced with SM for another 4 h. **b**, Anti-HA immunoblot showing Dam induction and depletion of Isw1 and Chd1. **c**, Nucleosome phasing changes as Rsc8 or Isw1 and Chd1 are depleted (for all time points, see Extended Data Fig. 7a). Methylation data obtained at each time point after SM addition for GATC sites in all genes are plotted relative to the +1 nucleosome dyad (smoothed with a 21 bp window). The gray profiles show dyad distributions for nuclei from wild-type (left), Rsc8-depleted (central) and *isw1Δ chd1Δ* (right) cells (MNase-seq data from refs. 26,49 normalized to 0.1). **d**, Selected phasing profiles for ease of comparison. Rsc8 depletion (left): the first (when auxin was added) and last (after Rsc8 depletion) time points. Isw1/Chd1 depletion (right): the first time point (when auxin was added) and time points with similar average methylation in wild-type and Isw1/Chd1-depleted cells. **e**, A diagram showing the effect of RSC depletion on global genic chromatin structure: the nucleosomal array is shifted toward the promoter, narrowing the promoter NDR. **f**, A diagram showing the effect of depleting both Isw1 and Chd1 on global genic chromatin structure: nucleosome spacing is disrupted, resulting in loss of phasing.

If correct, then the Dam rate at NDRs is determined by the slower NDR complex turnover rate, because the protein-free NDRs have already been methylated. In contrast, the M.SssI rate may be composed of two different rates, corresponding to protein-free NDRs and complexed NDRs, with an average that is faster than the Dam rate.

The median methylation at centromeric CG sites is much lower than at sites elsewhere in the genome, confirming that centromeres are generally inaccessible (Fig. 3c). We observed nucleosome phasing profiles similar to those obtained with Dam, but with much higher resolution, due to the much larger number of CG sites compared to

GATC sites (Fig. 3d). The rate of methylation at centromeric CG sites was ~14-fold lower than at ORF sites (Fig. 3e and Extended Data Fig. 5a–e). We infer that *CEN* nucleosomes are much more stable than canonical nucleosomes in living cells; slow methylation might reflect transient exposure of centromeric DNA during DNA replication[34].

Our yeast strain is mating type *a* (*MATa*), possessing an active *MATa* locus and an identical but silenced copy at the *HMRa* locus. Silencing depends on the Sir2, Sir3 and Sir4 proteins[35]. Our short-read data for Dam cannot distinguish between these copies. However, long nanopore reads contain unique flanking sequences, allowing us to distinguish *MATa* from *HMRa*. There is also a silenced copy of *MATα* at *HMLα*. Analysis of CG methylation in SIR-repressed heterochromatin shows that both silenced loci are methylated much more slowly than the active *MATa* locus, indicating that SIR-mediated silencing reduces accessibility in living cells (Fig. 3f and Extended Data Fig. 5b), consistent with early single-gene studies using Dam[36,37] and nuclei studies with restriction enzymes[35].

## Dynamic versus static chromatin

Our observation that most of the genome is protected in nuclei, particularly over coding regions where nucleosomes reside, indicates that nucleosomes protect DNA from methylation by Dam. So how does Dam access nucleosomal DNA in vivo? We propose that nucleosomes are static in nuclei, such that only linkers get methylated, but globally dynamic in living cells (Fig. 2i). Nucleosomes may be static in nuclei because crucial cofactors are absent (for example, ATP). In vivo, transient exposure of nucleosomal DNA to Dam may occur as a result of (1) core histone removal and replacement, (2) nucleosome mobility (sliding) and/or (3) reversible nucleosome conformational changes (Fig. 2i). Any or all of these potential sources of nucleosome dynamics may contribute to global accessibility. We expect these structural transitions to be transient and reversible, because phasing is generally preserved (Figs. 2f and 3d), indicating that nucleosomes retain their average positions even as their DNA is methylated, presumably oscillating between exposed and protected structures (via removal and replacement, or sliding back and forth, or opening and closing; Fig. 2i).

## Remodelers contribute to nucleosome flux

Yeast has several ATP-dependent chromatin remodelers possessing some or all of the proposed nucleosome-mobilizing activities (Fig. 2i): enzymes with sliding activity in vitro include CHD1, ISW1a/b, ISW2 and INO80 (refs. 38–41), although only CHD1, ISW1b and INO80 contribute to global nucleosome spacing in vivo[26,42–44]. SWI/SNF, RSC and INO80 can transfer histone octamers from one DNA molecule to another, and induce conformational changes, as well as slide nucleosomes[12]. CHD1, ISW1b, RSC and INO80 all have genome-wide effects on chromatin organization, as measured in nuclei by MNase-seq[26,42–48]. MNase-seq studies show that depletion of an essential RSC subunit results in a global shift of genic nucleosomes toward the promoter, narrowing the NDR[45,47–50], and that the *chd1Δ isw1Δ* double mutant has globally disrupted chromatin[26,43]. Both of these phenotypes imply that major net nucleosome movements have occurred in response to loss of the remodeler(s).

We tested whether we could detect net nucleosome movements in living cells as they are depleted of Rsc8, or of both Isw1 and Chd1, by employing the same auxin-dependent degron used for Dam. Cells were grown in the presence of auxin for 4 h to deplete the remodeler(s) and then SM was added for a time course up to 4 h in the continued presence of auxin. Wild-type cells were treated identically as a control. The endogenous genes encoding remodeler subunits Rsc8, Isw1 and Chd1 were all tagged with the same three HA tags used to detect Dam, allowing comparison of their relative levels in the same immunoblot (Fig. 4a,b). The Dam concentration in wild-type cells is much lower than that of Rsc8 before auxin is added, indicating that Dam is not expressed at high levels in wild-type cells (Fig. 4a). In Rsc8–degron

cells, Dam is induced with somewhat different kinetics and to lower levels than in the wild type, as Rsc8 is eventually depleted to ~5% of its initial level, potentially accounting for the slower methylation rate (Extended Data Fig. 6a–c). In Isw1/Chd1 double degron cells, Dam is induced to approximately threefold higher levels than the wild type, resulting in faster methylation, which reaches almost 100% (Extended Data Fig. 6b,c); Chd1-3HA disappears almost immediately, whereas Isw1-3HA depletion takes much longer, perhaps because there is more Isw1 than Chd1 to begin with (Fig. 4b).

Nucleosome phasing plots reveal that major net nucleosome movements occur during remodeler depletion (Fig. 4c). This is clear from overlays of data at zero time (just before auxin is added), when wild-type and remodeler–degron cells give almost identical phasing plots, and at the final time point after SM induction, by which time the phasing patterns have diverged (Fig. 4d and Extended Data Fig. 7a,b). In Rsc8–degron cells, there is a global shift of nucleosomes toward the promoter as Rsc8 is depleted (Fig. 4d,e), indicating that RSC contributes to chromatin flux by maintaining average nucleosome positions in wild-type cells, and that the net contribution of other factors, such as other remodelers, is antagonistic to RSC, promoting net movement toward the promoter. Although the +1 nucleosome shift is small (~24 bp), it is similar to that observed by MNase-seq (~17 bp (ref. 45)). In Isw1/Chd1 degron cells, phasing becomes increasingly poor with time, such that the +1 nucleosome peak is the only strong peak that persists (Fig. 4d). This indicates a transition from order to disorder, involving net nucleosome movement from their average wild type positions to apparently random positions downstream of the +1 nucleosome, as the spacing enzymes are depleted (Fig. 4f). These observations indicate that nucleosomes are globally mobile in living cells, since nucleosomal DNA becomes more and more methylated over time, coinciding with a net change in average nucleosome positions.

## Global genome accessibility

Our data indicate that chromatin is in a state of continuous flux in living yeast cells, unlike in nuclei, in which nucleosomes are static and stable, such that only linkers and the outer ~10 bp on each side of the nucleosome are accessible[20]. This flux may involve histone octamer displacement and rebinding, nucleosome sliding and/or nucleosome conformational changes (Fig. 2i). Histone turnover studies in yeast imply that global nucleosome displacement and rebinding occur during transcription in vivo[51]. On the other hand, fluorescence recovery after photobleaching studies in cells of higher organisms are consistent with only low levels of histone exchange[52]. Similarly, single molecule tracking studies in yeast provide little evidence for widespread global histone exchange[53,54]. These studies are more consistent with nucleosome sliding or conformational changes. A simple conformational change model implies that the nucleosomes do not change their average positions, and so cannot account for the global nucleosome shifts that we observe as RSC or Isw1 and Chd1 are depleted, which are most easily explained by nucleosome sliding to new average positions. Nevertheless, chromatin flux most probably reflects contributions from all three potential sources of nucleosome dynamics.

Our data indicate that the entire genome is methylated in a few hours. Although this may seem slow, the cell-to-cell variation in Dam production timing must be taken into account, as well as the Dam concentration, which may be rate limiting. Indeed, the rate of repair of ultraviolet-induced DNA lesions in yeast by a constitutively overexpressed photolyase is very fast, on a seconds timescale, indicating that lesions in nucleosomal DNA are readily accessible in vivo; furthermore, consistent with our observations, lesions in a centromere (*CEN14*) are not repaired[55]. The methylation rate will also depend on the exposure time of nucleosomal DNA. If histones are being transiently removed and replaced, the exposure time will be determined by the rates of removal and replacement. For nucleosome sliding, the exposure time will depend on the rate of sliding and perhaps on the location of the site

within the nucleosome, since a site located near the dyad is less likely to be exposed because nucleosomes may collide with one another as they move back and forth, given the short linker length in yeast. For a conformational change, the exposure time will depend on the stability of the structurally altered nucleosome and its degree of residual protection; RSC- and SWI/SNF-remodeled nucleosomes are quite stable in vitro[56–58].

## Competing remodelers and nucleosome flux

In living cells, both Rsc8 and Isw1/Chd1 depletion result in gradual global nucleosome shifts to new average positions with time, showing that these enzymes contribute to chromatin flux. The endpoints are similar to chromatin organization determined by MNase-seq using nuclei isolated from cells after Rsc8 is completely depleted[45,47,50], or from *isw1Δ chd1Δ* double mutant cells[26,43]. Similarly, MNase-seq studies of nuclei from mammalian cells depleted of the SNF2H/SMARCA5 ATP-dependent chromatin remodeler show increased nucleosome spacing[59,60]. In addition to RSC, ISW1 and CHD1, other yeast chromatin remodelers (for example, SWI/SNF, INO80, SWR1 and ISW2) may contribute to nucleosome dynamics. We propose that chromatin flux is the result of competing remodeling activities, which continuously move nucleosomes to create alternative chromatin organizations with different average nucleosome positions. Thus, depletion of a remodeler such as RSC, Isw1 or Chd1 removes one of these activities, resulting in a new chromatin organization representing the balance of the remaining competing activities.

Even though MNase-seq data indicate that most promoters are almost completely depleted of nucleosomes, our Dam methylation data show that the median GATC site in a promoter NDR is completely protected in a fraction of nuclei (as we also observed using AluI[20]). We have proposed that this protection reflects the presence of stable bound transcription complexes analogous to the TFIIIB–TFIIIC RNA polymerase III transcription complex formed on tRNA genes[22,31]. These transcription complexes may act as barriers to nucleosome formation, resulting in phasing[61,62]. In our version of this model, RSC is required for correct positioning of the +1 nucleosome next to the barrier complex at the promoter, and then ISW1 and CHD1 space the downstream nucleosomes along the gene using the +1 nucleosome as a reference point[49]. Thus, in RSC-depleted cells, the narrow NDR suggests that the barrier is smaller, allowing the +1 nucleosome to encroach on the promoter, resulting in a general shift of the entire nucleosomal array toward the promoter, as ISW1 and CHD1 space nucleosomes relative to the new +1 position[63]. In living cells, unlike in nuclei, promoters and tRNA genes are virtually completely accessible, indicating that the putative barrier complexes are also in flux. If so, phasing between barriers would also be changing, because the reference nucleosome would be changing as barriers dissociate and re-associate.

## Implications for gene regulation

A widely accepted model of gene regulation is that chromatin is generally repressive in nature, blocking access to genomic DNA, unless a series of activating events occurs[19]. It is envisaged that promoters are blocked by nucleosomes and perhaps by the formation of higher-order chromatin structures and heterochromatin, preventing transcription factors from binding to their cognate sites and activating transcription. The chromatin block is relieved by remodeling activities, primarily ATP-dependent remodelers, which are recruited to the regulatory element to move or remove nucleosomes blocking binding sites. This model posits remodeler-assisted dynamic transitions in chromatin from one static state to another. However, the observed global accessibility of the yeast genome suggests that transient exposure resulting from chromatin flux will eventually allow transcription factors to bind to all of their cognate sites. That is, chromatin is essentially transparent, except at centromeres and silenced loci. Indeed, we have shown, using ChIP-seq, that Gcn4 can access high-affinity sites in gene bodies as well

as in promoters in vivo, often with functional consequences[24,64]. We propose that sequence-specific transcription factors are the primary drivers of gene regulation in yeast; they bind when a site is exposed during flux and activate or repress genes according to their concentrations, motif affinity and gene-specific feedback systems. We also note that pioneer factors would not have to wait for site exposure. In conclusion, we propose that virtually all of yeast chromatin is in a state of continuous flux in living cells, due to competing remodeling activities, suggesting that chromatin packaging is not generally repressive in yeast. The exceptions are the silenced loci (*HMRa* and *HMLα*) and the 16 centromeric nucleosomes (one on each chromosome), which together account for <0.1% of the yeast genome. We note that yeast chromatin is essentially equivalent to euchromatin in the cells of higher organisms (with the exception of the silenced loci). In higher organisms, repressed genes generally reside in facultative heterochromatin, which may resemble the yeast silenced loci and so may well have different flux properties from euchromatin.

## Online content

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

## Methods

### Plasmids

The Dam integration cassette (Fig. 2a) is a 7,279 bp NotI insert in plasmid p876, containing upstream *HO* flanking sequence, then the TIR1 gene with seven C-terminal Myc tags driven by the yeast *ADH1* promoter, with the 3′ untranslated region (UTR) from *ADH1*, two Tet repressor binding sites (which were not used), a truncated *CUP1* promoter with a single high-affinity Gcn4 site instead of the upstream activating sequence (UAS), the Dam–ORF fused to a mini-degron and three HA tags, followed by the 3′ UTR from *CUP1*, the KanMX gene and part of the *HO* ORF. Initially, we linked the Dam ORF to the strong copper-inducible *CUP1* promoter. Although this worked well, we discovered that copper reduces cell viability for some mutants. Consequently, we replaced the proximal UAS in the *CUP1* promoter (which confers Cu dependence) with a single high-affinity Gcn4 transcription factor binding site[24]. p876 was constructed as follows: plasmid HO-poly-KanMX4-HO (ATCC 87804) (p783) is an integration plasmid based on pUC21 containing the KanMX selection marker flanked by regions from the *HO* gene[65]. The 2,387 bp PacI–AscI fragment containing the *CUP1* promoter, the yeast codon-optimized ORF for the M.AluI methyltransferase, a C-terminal HA tag and the *CUP1* 3′ UTR from p786 (constructed by gene synthesis based on pMK-RQ-Bs; Thermo Fisher GeneArt) was inserted at the PacI/AscI sites in p783 to obtain p791. The M.AluI-HA ORF in p791 was replaced with a yeast codon-optimized Dam ORF with a mini-degron and three C-terminal HA tags by substituting the 1,670 bp MfeI–AgeI fragment with the 1,140 bp MfeI–AgeI fragment containing the *CUP1* promoter fused to the Dam ORF, an auxin-dependent mini-degron (auxin-inducible degron (AID*); residues 71–114, ref. [66]) and two HA tags from p801, which was obtained by gene synthesis (Thermo Fisher GeneArt), resulting in p803. The TIR1 gene was obtained as a 2,714 bp SmaI fragment by polymerase chain reaction (PCR) using primers 2162 (5′-GACGTTCCCGGGCGCATGCAACTTCTTTTCTTT) and 2163 (5′-GACGTTCCCGGGCATATTACCCTGTTATCCCTAG) with pIS385-OsTIR1-7x_cMyc (p799) as template[67] and inserted at the SmaI site in p803, in the same orientation as the Dam ORF, to obtain p810. To correct an inactivating mutation (V223E) in the Dam ORF and to remove an AgeI site by changing the P222 codon from CCG to CCT, the 816 bp SnaBI–AgeI fragment in p810 was replaced with an 816 bp SnaBI–AgeI fragment made by PCR using primers 2209 (5′-CGAGCGTCATATACCTGTATTAATTAGCA), 2212 (5′-AGAACCGACGAGTACGTACAG) and 2215 (5′-AGACTATTCGTTTCAGCTAGCGT) to obtain p821. Two tetO sites were introduced into p821 and the *CUP1* promoter was truncated at the upstream EcoRV site by replacing the 1,043 bp PacI–SwaI fragment with the 961 bp PacI–SwaI fragment from p834 (obtained by gene synthesis based on pMK-RQ; Thermo Fisher GeneArt) to obtain p835. The *CUP1* promoter was further truncated to include only the proximal UAS by replacing the 529 bp AvrII–SwaI *CUP1* promoter fragment in p835 with a 435 bp AvrII–SwaI fragment obtained by PCR using primers 2223 (5′-GACGTTCCTAGGCGATGCGTCTTTTCCGCT) and 2224 (5′-GAATGCTCTATTCTTCTTCATTTAAATGTG) with p810 as template, to obtain p840. Finally, p876 was constructed by substituting the 216 bp AvrII–SwaI *CUP1* promoter fragment in p840 with a 193 bp AvrII–SwaI *CUP1* promoter fragment, in which the UAS was replaced by a single high-affinity Gcn4 site (ATGACTCAT[24]), obtained by PCR using primers 2239 (5′-phos-AAATGTGATGATTGATTGATTGATTGTACA) and 2312 (5′-GACGTTCCTAGGGCATGCGATGACTCATCTACCAACGCAATATGGATTGTC). This step converted the copper-dependent promoter to an SM-inducible promoter. To obtain an integration plasmid for Dam–degron–GFP-3HA (p887; Extended Data Fig. 4a), a 990 bp AfeI–NheI fragment containing the degron–spacer–GFP-3HA coding sequence was synthesized (p852; Thermo Fisher GeneArt) and used to replace the 236 bp AfeI–NheI fragment in p840 to obtain p854 (the GFP sequence is the same as in pYM44 (ref. [68])). Then, p887 was constructed by replacing the 216 bp AvrII–SwaI *CUP1* promoter fragment in p854

with the 193 bp AvrII–SwaI Gcn4 site promoter fragment, as above. Plasmid p906 containing the M.SssI integration cassette (Fig. 3a) was constructed by replacing the 1,031 bp SwaI–AfeI Dam ORF fragment in p876 with the 1,355 bp SwaI–AfeI yeast codon-optimized M.SssI ORF fragment from p905 (obtained by gene synthesis; Thermo Fisher GeneArt).

All plasmids designed to fuse the auxin-inducible degron and HA tags to the C-termini of remodeler subunits were constructed by gene synthesis (Thermo Fisher GeneArt) as a NotI insert, and include the 3′ end of the remodeler subunit ORF fused to the same degron used for Dam and three HA tags at the C-terminus, followed by 3′ genomic sequence with a selection marker inserted in the opposite orientation. For *RSC8*, p877 is based on pMA-RQ, with *LEU2* inserted downstream. For *ISW1*, p892 is based on pMA, with *URA3* downstream. For *CHD1*, p893 is based on pMK-RQ, with *LEU2* downstream. All plasmid sequences were verified and are available on request.

### Yeast strains

YHP808, YHP817 and YHP827 were obtained by transformation of YDC111 (*MATa ade2-1 can1-100 leu2-3,112 trp1-1 ura3-1* (ref. [69])) with a NotI digest of p876, p887 or p906, respectively, and selection for G418 resistance. Integration at the *HO* locus was confirmed by PCR. Rsc8 and Isw1/Chd1 degron strains (YHP812 and YHP823, respectively) were obtained by transformation of YHP808 with NotI digests of p877, or both p892 and p893 (sequentially), respectively, and appropriate selection. Integration at the correct locus was confirmed by PCR. All yeast strains are available on request.

### Dam methylation of yeast nuclei

Nuclei were prepared from YDC111 cells as described previously[20]. Pellets of nuclei from ~100 $A_{600}$ units of cells were resuspended in 4 ml prewarmed Dam methylase low ionic strength buffer (10 mM HEPES-K pH 7.5, 35 mM NaCl, 5 mM $MgCl_2$, 80 µM *S*-adenosylmethionine (SAM) and divided into five 400-µl aliquots. Dam methyltransferase (New England Biolabs (NEB) M0222L, 8 units per µl; 1.6 µg Dam per ml) was added (0, 50, 100, 200 or 400 units, final concentration in nM 0, 0.77, 1.5, 2.9, 5.6) to the nuclei, mixed gently and incubated for 20 min at 25 °C. Methylation reactions were stopped by adding 90 mM EDTA, 9% sodium dodecyl sulfate (SDS) to final concentrations of 10 mM and 1%, respectively. DNA was extracted by the addition of 20% SDS to 1% with mixing and the addition of 5 M potassium acetate to 1 M, followed by two extractions with an equal volume of chloroform using Phase Lock Gel (Heavy; Quanta Bio 2302830), precipitation with 0.7 vol isopropanol, and one wash with 70% ethanol. The purified DNA was dissolved in 50 µl 10 mM Tris–HCl pH 8.0, 0.1 mM EDTA, 0.4 mg ml⁻¹ RNase A and incubated at 37 °C for 1 h. Genomic DNA was further purified using 0.8 vol Ampure XP beads to DNA solution (Beckman-Coulter, A63881) and eluted in 45 µl 10 mM Tris–HCl pH 8.0, 0.1 mM EDTA (TE (0.1)). The purified DNA was digested with 10 units DpnI (NEB R0176L; 20 U µl⁻¹) at 37 °C for 1.5 h. The reaction was stopped by adding EDTA to 10 mM and SDS to 0.1%. The DNA was analyzed in a 1% (wt/vol) agarose gel stained with SYBR Gold (Invitrogen, S11494). DpnI-digested DNA was purified using 1.8 vol AMPure XP beads and eluted in 50 µl TE (0.1). Any nicks were sealed by treating the DNA with 1 µl HiFi Taq DNA ligase (NEB, M0647S) in the buffer provided at 37 °C for 1 h. The reaction was stopped by adding EDTA to 20 mM. The DNA was transferred to a sonicator tube (Covaris, 520166) and fragmented using a Covaris ME220 ultrasonicator at 4 °C (350 bp program: peak power, 50 W; duty factor 10%; 1,000 cycles per burst; average power 5; total time of 150 s per tube). The DNA fragment size was checked by analysis of 5 µl DNA in a 2% (wt/vol) agarose gel stained with SYBR Gold (the DNA size ranged from ~100 to ~700 bp) using MassRuler DNA Ladder Mix (Thermo Fisher, SM0403) as a marker. The remaining DNA was purified using 1.8 vol AMPure XP beads and eluted in 50 µl TE (0.1). DNA concentrations were determined by measuring at $A_{260}$. Illumina paired-end libraries

were prepared as described previously[20], using the end-repair (NEB, E7371AA) and NEBNext ultra-DNA library kits (NEB, E7370). Sequencing was performed using an Illumina NextSeq-500 machine.

## Dam methylation in living cells

YHP808 and YHP817 were grown in 250 ml synthetic complete medium without isoleucine or valine (SC−ile−val) with 1 mM auxin (Sigma, I2886) at 30 °C to optical density (OD)$_{600}$ of ~0.7–0.8. For the 0 min time point, 5 and 10–15 OD units of cells were removed for protein and DNA extraction, respectively. Auxin was removed from the remaining culture, by washing the cells by filtration with twice the volume of fresh medium. The cells were resuspended in 225 ml fresh medium and SM (Sigma, 34224; 2 mg ml$^{-1}$ dimethyl sulfoxide) was added to a final concentration of 1 µg ml$^{-1}$ SM to induce Dam production. Aliquots of cells were removed after 30, 60, 120 and 240 min at 30 °C for genomic DNA (10–15 OD units) and protein extraction (5 OD units), spun down at 4 °C and stored at −80 °C. For DNA extraction, cells were extracted three times with 50 mM Tris pH 8.0, 5 mM EDTA (5× TE buffer) with 2% SDS, to remove cell contents, while retaining the cell wall. SDS was removed by washing the cells three times with 5× TE. Genomic DNA (450 µl) was released by digestion with 50 µl lyticase (Sigma, L2524 at 25,000 U ml$^{-1}$) for up to 10 min at 37 °C. Digestion of the cell wall was monitored by measuring the OD$_{600}$ of 10 µl cell suspension in 1 ml 1% SDS and considered complete when the OD$_{600}$ decreased to <10% of the initial value. The reaction was stopped with 50 µl 20% SDS and DNA was extracted by adding 110 µl of 5 M potassium acetate, followed by two extractions with an equal volume of chloroform, precipitation with 0.7 vol isopropanol and one wash with 70% ethanol. About 2 µg genomic DNA was digested with 10 units DpnI (NEB, R0176L; 20 U µl$^{-1}$) in NEB CutSmart buffer for 1.5 h at 37 °C. The DNA was purified as described above for nuclei. For cell cycle arrest, YHP808 was grown in SC−ile−val medium with 1 mM auxin as above to an OD$_{600}$ of 0.3–0.4 and treated with α-factor at 10 µg ml$^{-1}$ (FDA Core Facility) for 3 h. The cells were washed as above to remove auxin and resuspended in SC−ile−val medium with α-factor and induced with SM as above. Cell aliquots were removed as above. For experiments with remodeler mutants and the wild-type control, cells were grown in SC−ile−val and auxin was added to 1 mM for 4 h to deplete the remodeler protein and reduce the Dam background, followed by a 4 h time course of SM treatment in the presence of auxin. Cell aliquots were removed before auxin addition, after 2 h, 3 h and 4 h of auxin addition, and after 30, 60, 120 and 240 min of SM treatment.

## Immunoblotting

Cell pellets corresponding to 5 OD units were resuspended in 100 µl lithium dodecyl sulfate buffer with 0.2 M 2-mercaptoethanol and heated for 5 min at 99 °C. Solids were removed by centrifugation at 9,400g for 2 min and 2 µl of supernatant was loaded in two 4–12% Bis–Tris NuPAGE gels (Thermo Fisher) and run using NuPAGE MOPS SDS running buffer. One gel was stained with Coomassie blue; the other was transferred to a polyvinylidene difluoride membrane using an iBlot machine (Invitrogen) according to the manufacturer's instructions. After transfer, the membrane was blocked using 10–20 ml 5% skim milk in TBS (20 mM Tris–HCl, pH 8.0 and 0.5 M NaCl) with 0.1% Tween-20 for 1 h at room temperature with rotation. The membrane was incubated with horseradish peroxidase-conjugated anti-HA antibody (3F10; Roche, 12013819001) at 1:5,000 dilution overnight at 4 °C in blocking buffer. The membrane was washed at least three times with 10 ml TBS/0.1% Tween-20 at room temperature for 10 min with rotation. The membrane was treated with SuperSignal West Pico Plus chemiluminescent substrate (Thermo Fisher, 34580) for 5 min using a 1:1 mix of enhancer and peroxidase solution, and then excess solution was removed. The membrane was exposed to film (Amersham Hyperfilm ECL). To detect tubulin, the membrane was washed once with phosphate-buffered saline (PBS)/0.1% Tween-20 to remove the anti-HA signal, and blocked again with 5% skim milk in PBS/0.1% Tween-20 for

1 h at room temperature with rotation. The membrane was incubated with horseradish peroxidase-conjugated antitubulin antibody (Abcam, ab-185067) diluted 1:20,000 in PBS/0.1% Tween-20 with 0.5% skim milk for 1 h at room temperature with rotation, washed at least three times with PBS/0.1% Tween-20 and then exposed to film as above.

## M.SssI methylation and nanopore sequencing

YHP827 or YDC111 (negative control) cells were grown in auxin and induced with SM as above. Genomic DNA was purified using the Monarch High Molecular Weight Extraction kit (NEB, T3050/T3060) following the protocol for yeast, except that cell walls were removed using lyticase, as described above for at least 30 min at 37 °C; digestion was followed as described above. DNA samples (1–1.5 µg) were prepared for nanopore sequencing using the Ligation Sequencing Kit (SQK-LSK110, Oxford Nanopore Technologies (ONT)). Barcoding was performed using the Native Barcoding Kit (ONT, SQK-LSK109 and EXP-NBD104). Sequencing was performed using revision 9.4.1 flow cells on a MinION Mk1C instrument with the latest MinKNOW software updates (ONT). Half of each DNA sample was loaded onto a flow cell and sequenced for 20–24 h, then the flow cell was washed with DNase I (ONT Flow Cell Wash Kit EXP-WSH004) to clear any clogged pores, the DNase I was removed and the remaining half of the sample was loaded and sequenced for an additional 20–24 h. FAST5 files generated by the MinION instrument were base called with Guppy. Base-called reads were indexed using Nanopolish software v.0.14.0 (ref. [70]). Reads were mapped to sacCer3 with Minimap2 v.2.24 (ref. [71]) and sorted and indexed with samtools v.1.17 (ref. [72]). Reads were scored for methylation using Nanopolish[70]. Some CG sites are grouped by Nanopolish, depending on their proximity, resulting in ~260,000 CG sites (varying slightly from one data set to the next). MATLAB scripts were used to analyze the Nanopolish Excel file output. CG sites with very low coverage (<10% of the median coverage) were filtered out. For analysis of the *MATa*, *HMRa* and *HMLα* loci, we constructed a reference genome for our *MATa* strain, because the sacCer3 genome is *MATα*. The YHP827 genome was assembled using Canu v.2.1 (ref. [73]) with all Nanopore sequences from replicate 2 and polished using Medaka v.1.7.2 (ONT). The genome was polished further using Freebayes[74] and the Illumina sequences obtained from untreated wild-type nuclei (samples GSM7177980 and GSM 7177985 in GSE230309). Contigs were aligned to the sacCer3 genome using blastn to assemble all chromosomes from telomere to telomere, except for chromosome XII, which is interrupted by the ribosomal DNA locus (left half of 'XII_1' and right half =of 'XII_2'). Mating loci were annotated manually and defined as the region between the stop codons of each divergent gene pair (*a1* and *a2*, or *α1* and *α2*). The YHP827 genome sequence is available on GitHub.

## Bioinformatics and data analysis

Scripts were adapted from our AluI study[20]. Paired-end reads (50 nt) were aligned to the *Saccharomyces cerevisiae* reference genome sacCer3 using Bowtie 2 (ref. [75]) with the parameters -X 5000 -very-sensitive, to map sequences up to 5 kb with maximum accuracy. We counted the number of times each genomic base pair is contained in a sequenced DNA fragment to obtain the occupancy (fragment coverage) for each genomic base pair. We also counted the number of times each genomic base pair is the first (left end) or the last (right end) base pair in each sequenced DNA fragment. DpnI gives blunt ends, so if a Dam/DpnI site (GATC) is cut, two DNA fragments are generated, one ending with GA and the other beginning with TC. However, we observed that a substantial fraction of ends were consistent with the removal of the A (ending with the G) or removal of the T (beginning with the C). Presumably, the terminal A or T is sometimes removed during library preparation or perhaps by DpnI. Consequently, for the right end, the fraction cut by DpnI was calculated as the number of DNA fragments ending on A plus the number of DNA fragments ending on G divided by the fragment coverage of the G. Similarly, for the left end, the fraction cut by DpnI

# Article

was calculated as the number of DNA fragments ending on T plus the number of DNA fragments ending on C divided by the coverage of the C. Thus, two values for 'fcut' are obtained, one for the right end and one for the left end of each GATC site. These values were considered separately in the analysis, unless the GATC site in question is within 200 bp of a neighboring site. Since fragments <200 bp are recovered inefficiently (evident from lower coverage of such fragments in heavily digested samples), data for GATC sites with neighboring sites closer than 200 bp on both sides were excluded from the analysis, whereas data for GATC sites with a neighboring site closer than 200 bp only on one side were adjusted such that 'fcut' for the side with the close neighboring site was set equal to 'fcut' for the other side. In individual site plots, the average 'fcut' for the left and right ends was used. A table is provided with all 35,830 GATC sites listed with their location, site classification, genomic region and transcription decile. This table was used in MATLAB scripts for median and phasing analyses.

## Statistics and reproducibility
Two biological replicate experiments were performed (for Pearson correlations for fcut values, see Extended Data Table 1). The panels shown in each main figure belong to the same experiment. Extended data figures generally show the results from both replicate experiments, except for immunoblots, micrographs and DNA gels. The immunoblots, micrographs and DNA gel analyses were similar in both experiments.

## Reporting summary
Further information on research design is available in the Nature Portfolio Reporting Summary linked to this article.

## Data availability
The Illumina and Nanopore sequence data described in this paper are publicly available at the Gene Expression Omnibus database under accession number https://www.ncbi.nlm.nih.gov/geo/query/acc.cgi?acc=GSE230309. The sacCer3 genome sequence is available at the Saccharomyces Genome Database. Source data are provided with this paper.

## Code availability
The code used to analyze the Illumina and nanopore data is publicly available at GitHub: https://github.com/clarkda24/Prajapati_et_al_2023.git.

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

## Acknowledgements
We thank A. Hinnebusch (Eunice Kennedy-Shriver National Institute of Child Health and Human Development; NICHD) for helpful comments on the manuscript and T. Karpova (National Cancer Institute) for useful discussion. We thank the Hinnebusch laboratory for p799 and the Cassellas laboratory (National Institute of Arthritis and Musculoskeletal and Skin Diseases) for p783. This study utilized the high-performance computational capabilities of the Biowulf Linux cluster at the National Institutes of Health (NIH). This research was supported by the Intramural Research Program of the NIH (NICHD).

## Author contributions
H.K.P., P.R.E. and P.A.E. performed the Dam experiments. C.T.C., H.K.P. and Z.X. performed the Nanopore experiments and analysis. D.J.C. performed the bioinformatic analysis. H.K.P. and D.J.C. wrote the manuscript.

## Competing interests
The authors declare no competing interests.

## Additional information
**Extended data** is available for this paper at https://doi.org/10.1038/s41594-024-01318-2.

**Correspondence and requests for materials** should be addressed to David J. Clark.

## a

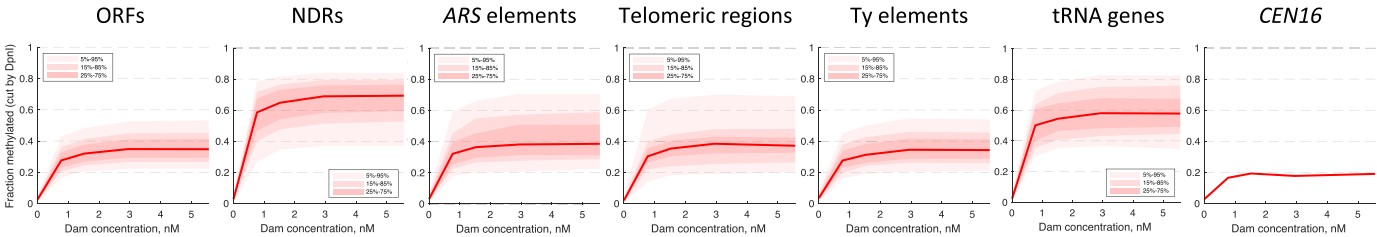

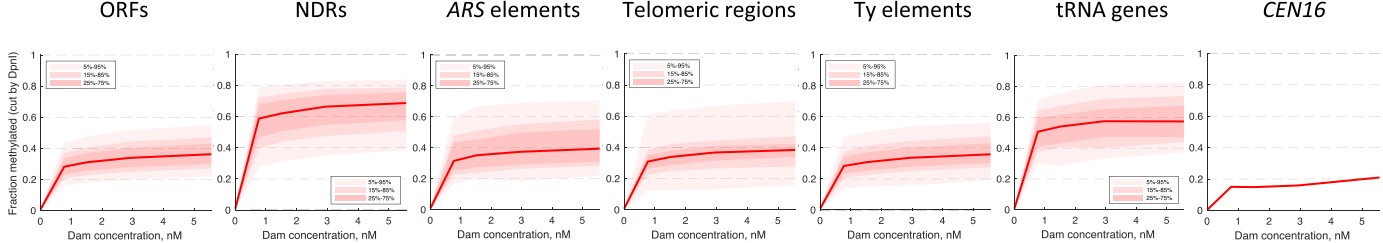

## b

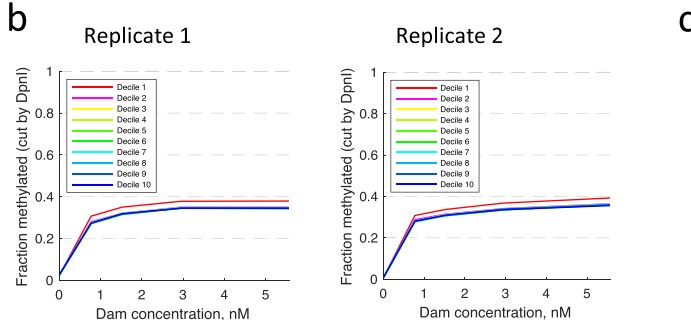

## c

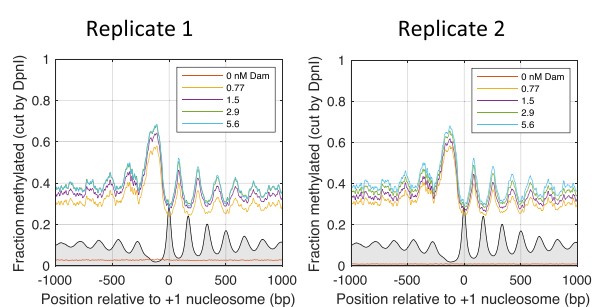

**Extended Data Fig. 1 | Measurement of DNA accessibility in nuclei.** Data for various genomic regions: biological replicates. **a**, Plots of median GATC site methylation against Dam concentration for different genomic regions. Red line and shading: median GATC site with data range indicated. *CEN16* has a single GATC site located at the edge of the centromeric nucleosome. **b**, Transcriptional activity has a marginal effect on median methylation of ORF sites. The ~5770 yeast genes were divided into deciles according to their RNA polymerase II levels (using ChIP-seq data for the Rpb3 subunit[29]). Data for the median GATC site in each decile are shown; the most active genes are in Decile 1. **c**, Nucleosome phasing detected by Dam methylation. Methylation data obtained at different Dam concentrations for GATC sites in all ~5,770 yeast genes are plotted relative to the +1 nucleosome dyad (smoothed with a 21-bp window). Grey profile: nucleosome dyad distribution (MNase-seq data[26] normalised to 0.1).

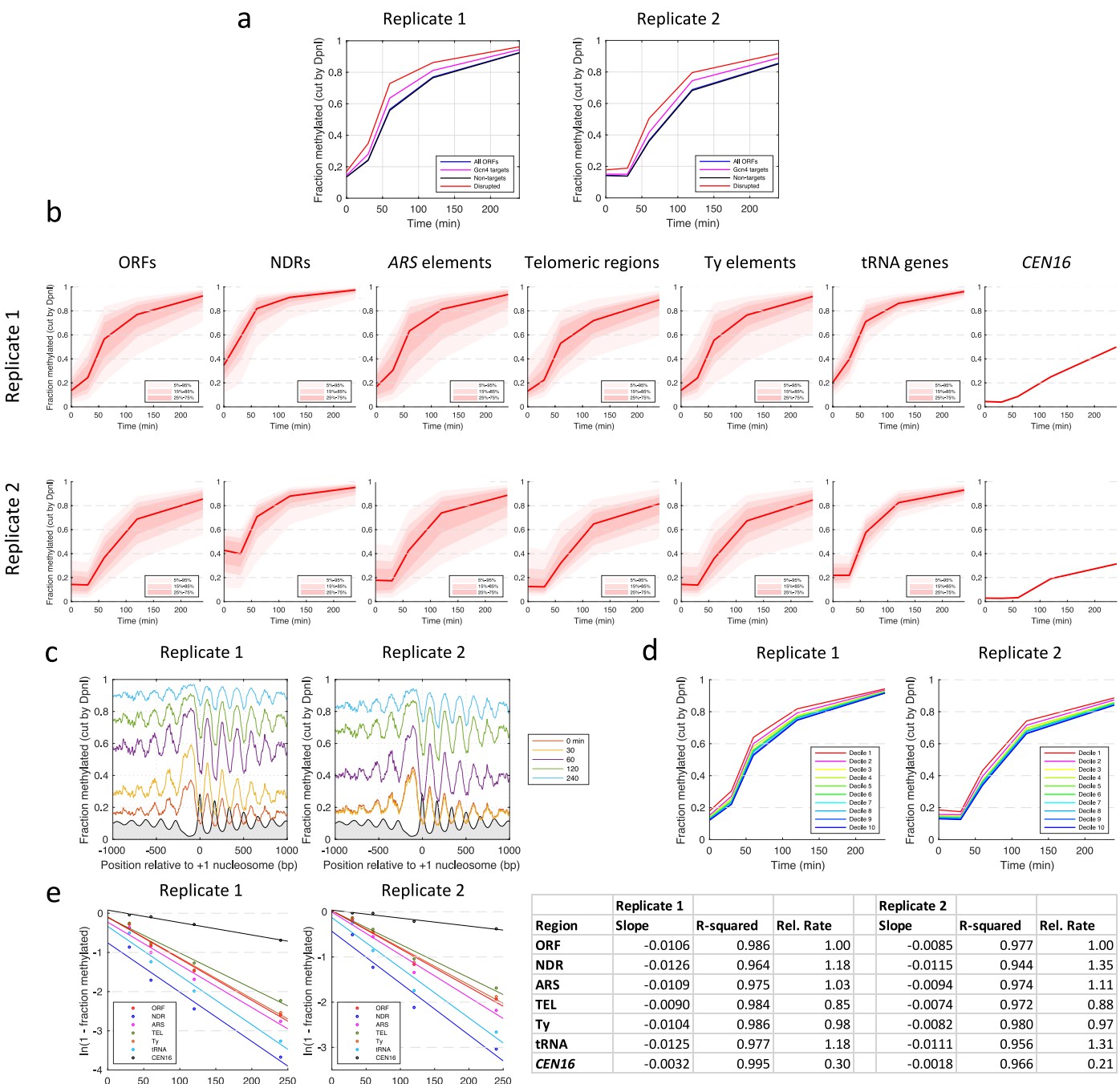

**Extended Data Fig. 2 | Dam methylation rates in various genomic regions in living cells. a**, Gcn4-activated genes are methylated faster than other genes. The 512 Gcn4-activated genes contain 2496 GATC sites[27]. A subset of 49 genes containing 235 GATC sites has disrupted chromatin correlated with high Pol II levels[28,29]. **b**, Almost complete methylation of GATC sites in all genomic regions except for the site in *CEN16*, which is methylated much more slowly. Red line and shading: median GATC site with data range indicated. **c**, Nucleosome phasing detected by Dam methylation: nucleosomal DNA is gradually almost fully methylated. Methylation data obtained for GATC sites in all genes at each time point are plotted relative to the +1 nucleosome dyad (smoothed with a 21-bp

window). Grey profile: nucleosome dyad distribution in nuclei (MNase-seq data normalised to 0.1). **d**, Transcriptional activity has a minor effect on methylation rate. The ~5770 yeast genes were divided into deciles according to their RNA polymerase II levels (using ChIP-seq data for the Rpb3 subunit[29]). Data for the median GATC site in each decile are shown; the most active genes are in Decile 1. **e**, Various genomic regions are methylated at quite similar rates, except for the GATC site in *CEN16*. Plot of the natural log of the median unmethylated fraction for GATC sites in each region against time after SM induction. Table: slopes (apparent rate constants), coefficients of determination ($R^2$) and rates relative to ORF sites.

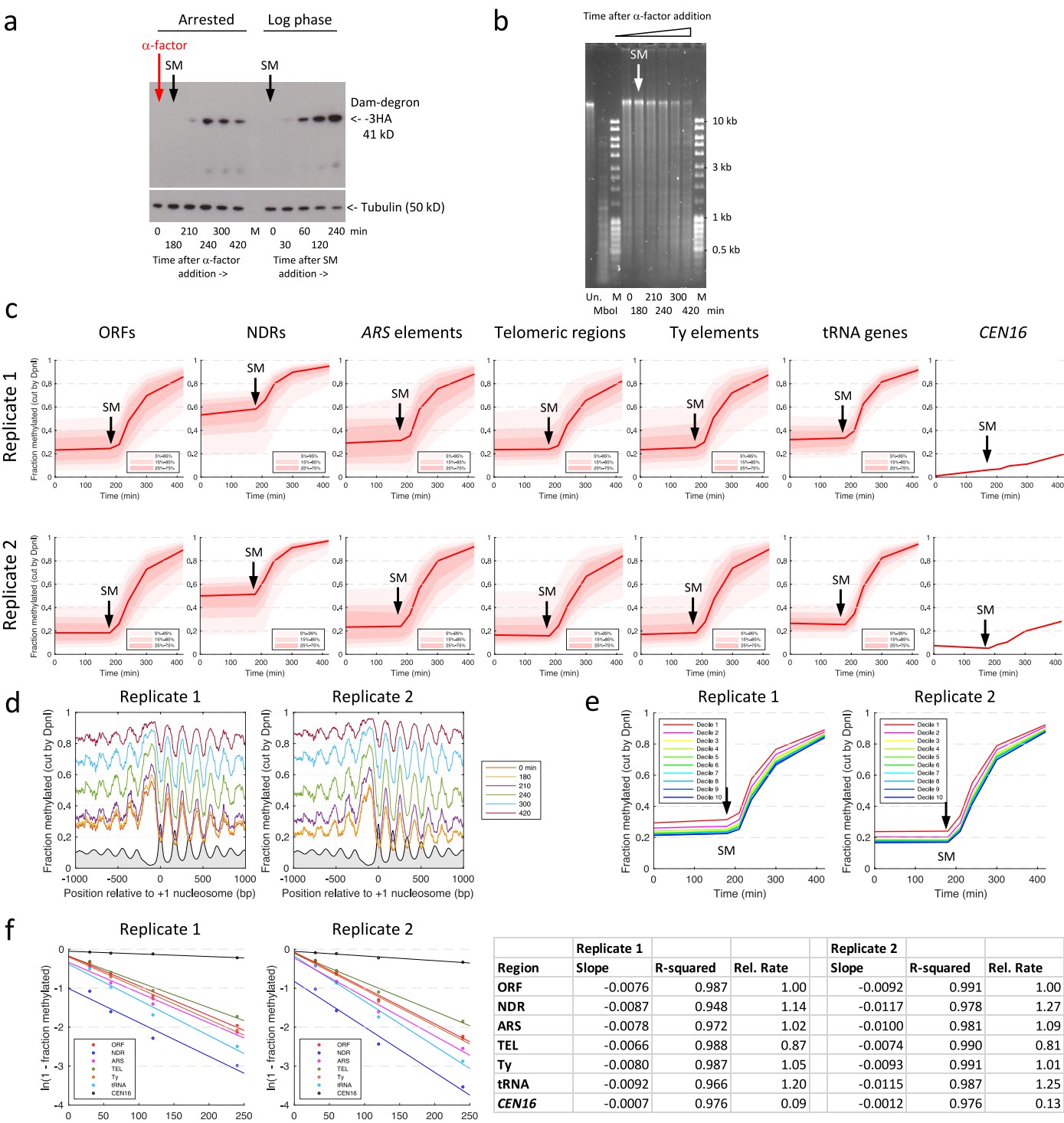

**Extended Data Fig. 3 | The genome is globally accessible in yeast cells arrested with α-factor. a**, Induction of Dam-degron-3HA by SM after arrest in G1. Cells grown in auxin were treated for 3 h with a high concentration of α-factor to prevent release, followed by addition of SM to induce Dam over a time course of 4 h. Anti-HA immunoblot comparing arrested cells with growing cells. **b**, Genomic DNA appears almost completely methylated after Dam induction in arrested cells. Electrophoretic analysis in a 1% agarose gel stained with SYBR-Gold. Un: undigested genomic DNA; MboI: unmethylated genomic DNA fully digested with MboI (which cuts unmethylated GATC). M: DNA size marker. **c**, Almost complete methylation of GATC sites in arrested cells in all genomic regions, except for the *CEN16* site. Red line and shading: median GATC site with data range indicated. **d**, Nucleosome phasing detected by Dam methylation in arrested cells: nucleosomal DNA is gradually methylated during the time course.

Methylation data obtained for GATC sites in all genes at each time point are plotted relative to the +1 nucleosome dyad (smoothed with a 21-bp window). Grey profile: nucleosome dyad distribution (MNase-seq data normalised to 0.1). **e**, Transcriptional activity has a minor effect on methylation rate in arrested cells. The ~5770 yeast genes were divided into deciles according to their RNA polymerase II levels (using ChIP-seq data for the Rpb3 subunit[29]). Data for the median GATC site in each decile are shown; the most active genes are in Decile 1. **f**, Various genomic regions are methylated at quite similar rates in arrested cells, except for the GATC site in *CEN16*. Plot of the natural log of the median unmethylated fraction for GATC sites in each region against time after SM induction. Table: slopes (apparent rate constants), coefficients of determination ($R^2$) and rates relative to ORF sites.

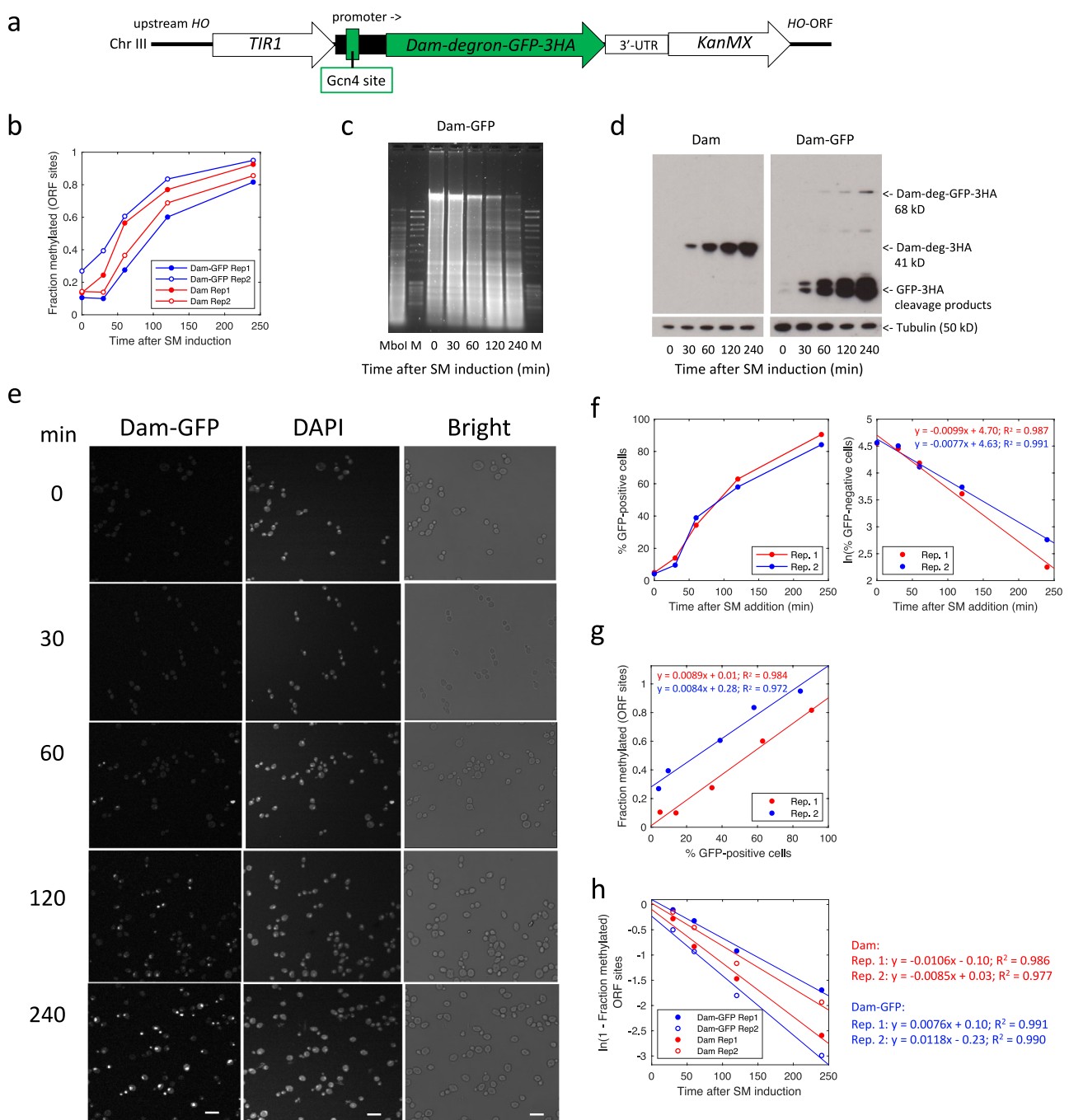

**Extended Data Fig. 4 | Cell-to-cell variation in Dam expression. a**, Integration construct used for SM-inducible Dam-degron-GFP-3HA fusion protein expression. **b**, Dam and Dam-GFP methylate GATC sites in ORFs at similar rates. **c**, Genomic DNA appears almost completely methylated after Dam-GFP induction. Electrophoretic analysis of DpnI-digested DNA in a 1% agarose gel stained with SYBR-Gold. MboI: unmethylated genomic DNA digested to completion with MboI. M: DNA size marker. **d**, Induction of Dam-degron-GFP-3HA by SM: cleavage of the fusion protein in vivo. Anti-HA immunoblot.

**e**, Cell-to-cell variation in induction of GFP fluorescence. Photomicrographs of yeast cells at each time point after SM induction: GFP fluorescence, DAPI stain and bright field. Scale bar = 10 μm. **f**, The fraction of GFP-negative cells decreases exponentially. Plot of % GFP positive cells against time after SM addition (left panel); exponential plot of the same data (right panel). **g**, The fraction of methylated ORF sites is directly proportional to the fraction of GFP-positive cells. **h**, The fraction of unmethylated ORF sites also decreases exponentially.

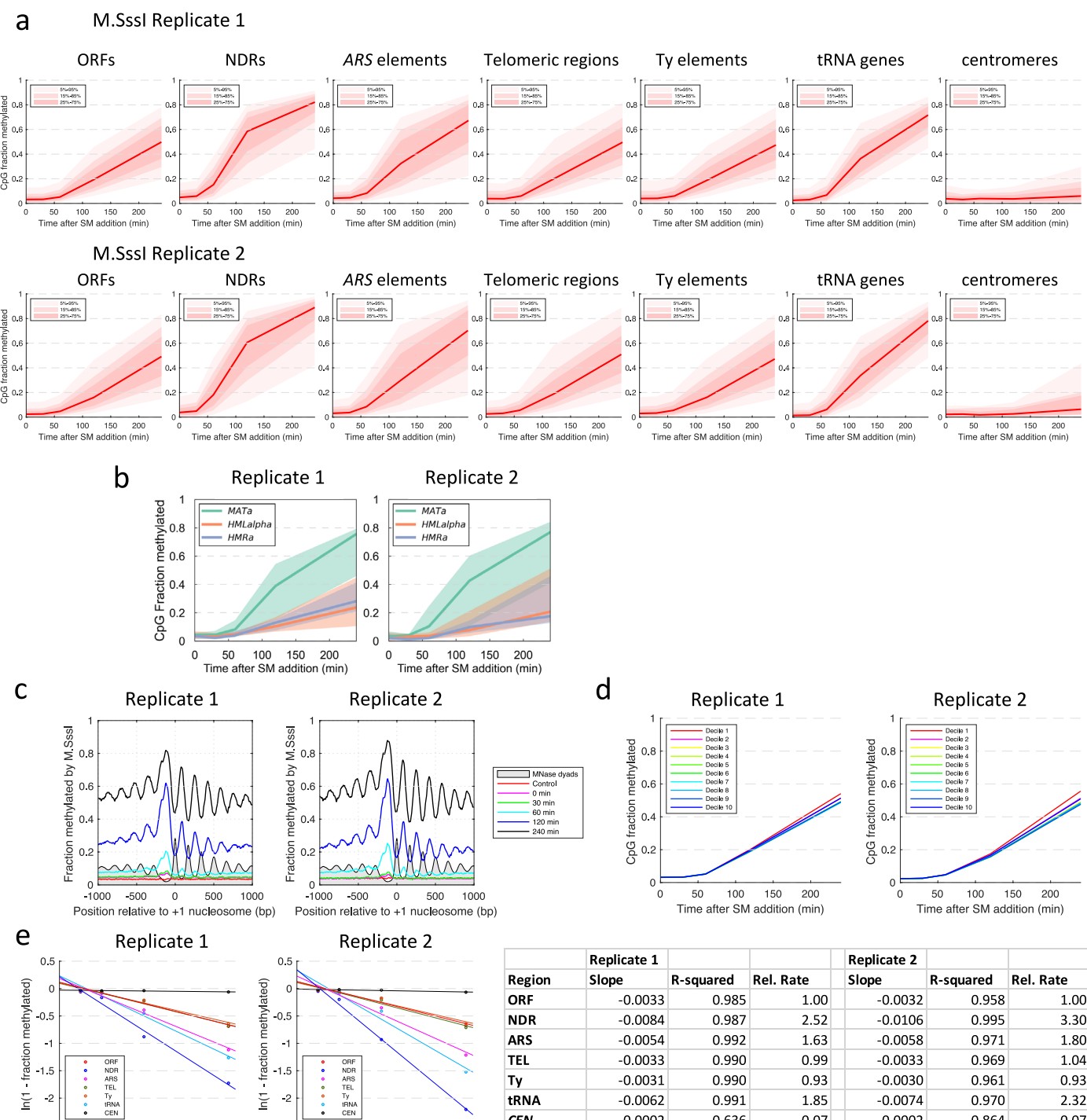

**Extended Data Fig. 5 | M.SssI methylation rates in different genomic regions in living cells.** Nanopore data for biological replicates. **a**, Centromeric CG sites are largely protected from methylation. Red line and shading: median CG site with data range indicated. **b**, Methylation of the silenced *HMLa* and *HMRa* loci is much slower than at the active *MATa* locus. Each line indicates the median CG site with the data range indicated by shading (25 to 75%). **c**, Nucleosome phasing detected by M.SssI methylation in living cells. Methylation data obtained for CG sites in all genes at each time point are plotted relative to the +1 nucleosome dyad (smoothed with a 21-bp window). Grey profile: nucleosome dyad distribution in nuclei (MNase-seq data normalised to 0.1). **d**, Transcriptional activity has a minor effect on methylation rate. The ~5770 yeast genes were divided into deciles according to their RNA polymerase II levels (using ChIP-seq data for the Rpb3 subunit[29]). Data for the median CG site in each decile are shown; the most active genes are in Decile 1. **e**, Methylation rates for various genomic regions. Plots of the natural log of the median unmethylated fraction for CG sites in each region as a function of time after SM induction. Table: rates relative to ORF sites: slopes and $R^2$ values for different regions for both replicates.

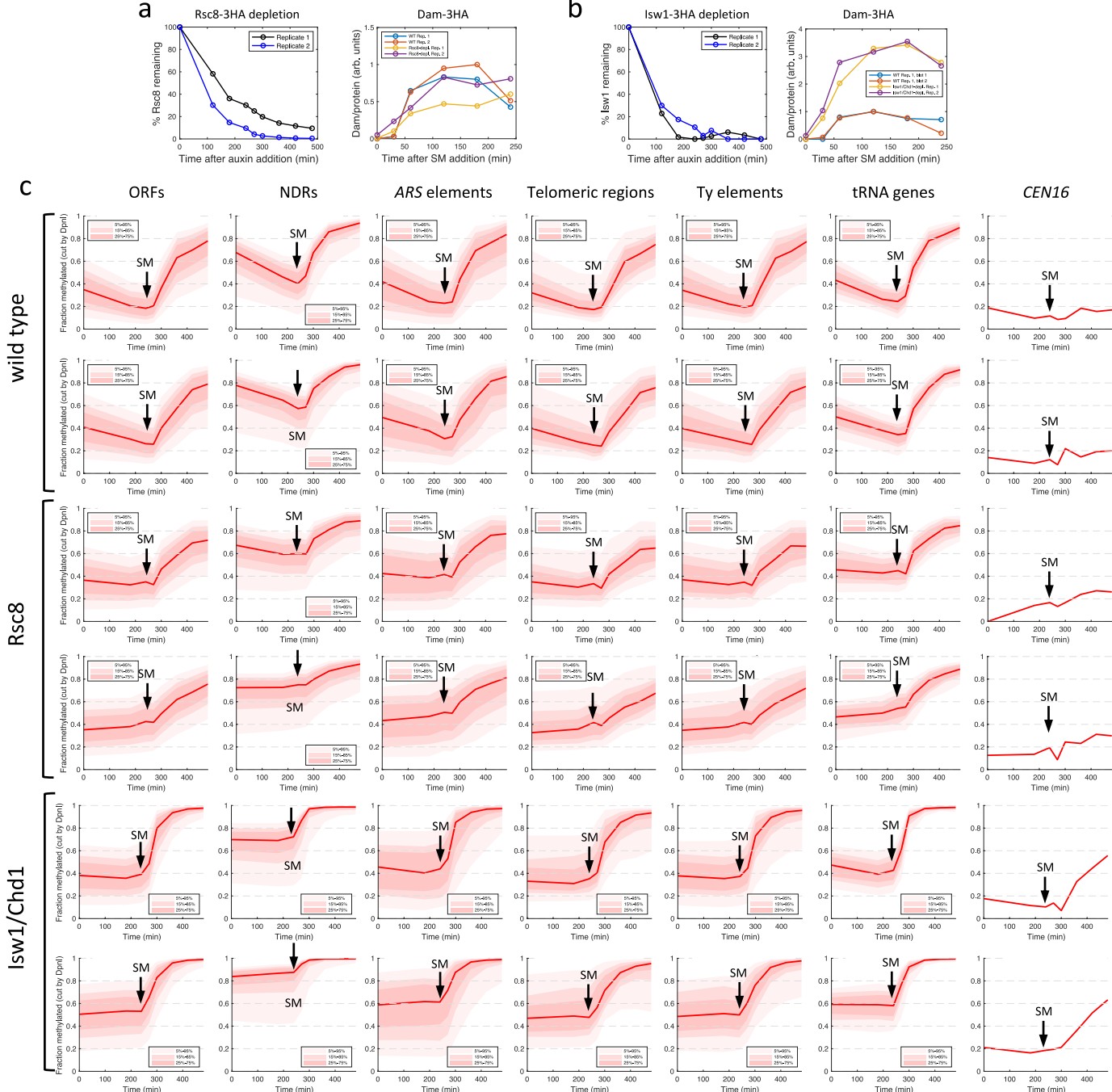

**Extended Data Fig. 6 | Dam methylation rates in cells depleted of Rsc8 or Isw1 and Chd1.** Cells were grown in auxin for 4 h and then SM was added for a 4 h time course. **a**, Time courses of Rsc8 depletion and Dam induction. Immunoblot data; error bars: standard deviation for two biological replicates (replicates were analysed in the same blot and normalised to total protein). The maximum wild type Dam-3HA signal was set at 1. **b**, Time courses of Isw1 depletion (Chd1 is absent by the next time point after auxin addition) and Dam induction.

Immunoblot data; error bars: standard deviation for two biological replicates (replicates were analysed in different blots relative to wild type replicate 1 and normalised to total protein). The maximum wild type Dam-3HA signal for replicate 1 in each blot was set at 1 (wild type data from both blots are shown). **c**, Methylation of GATC sites in various genomic regions in cells depleted of Rsc8 or Isw1 and Chd1. Red line and shading: median GATC site with data range indicated.

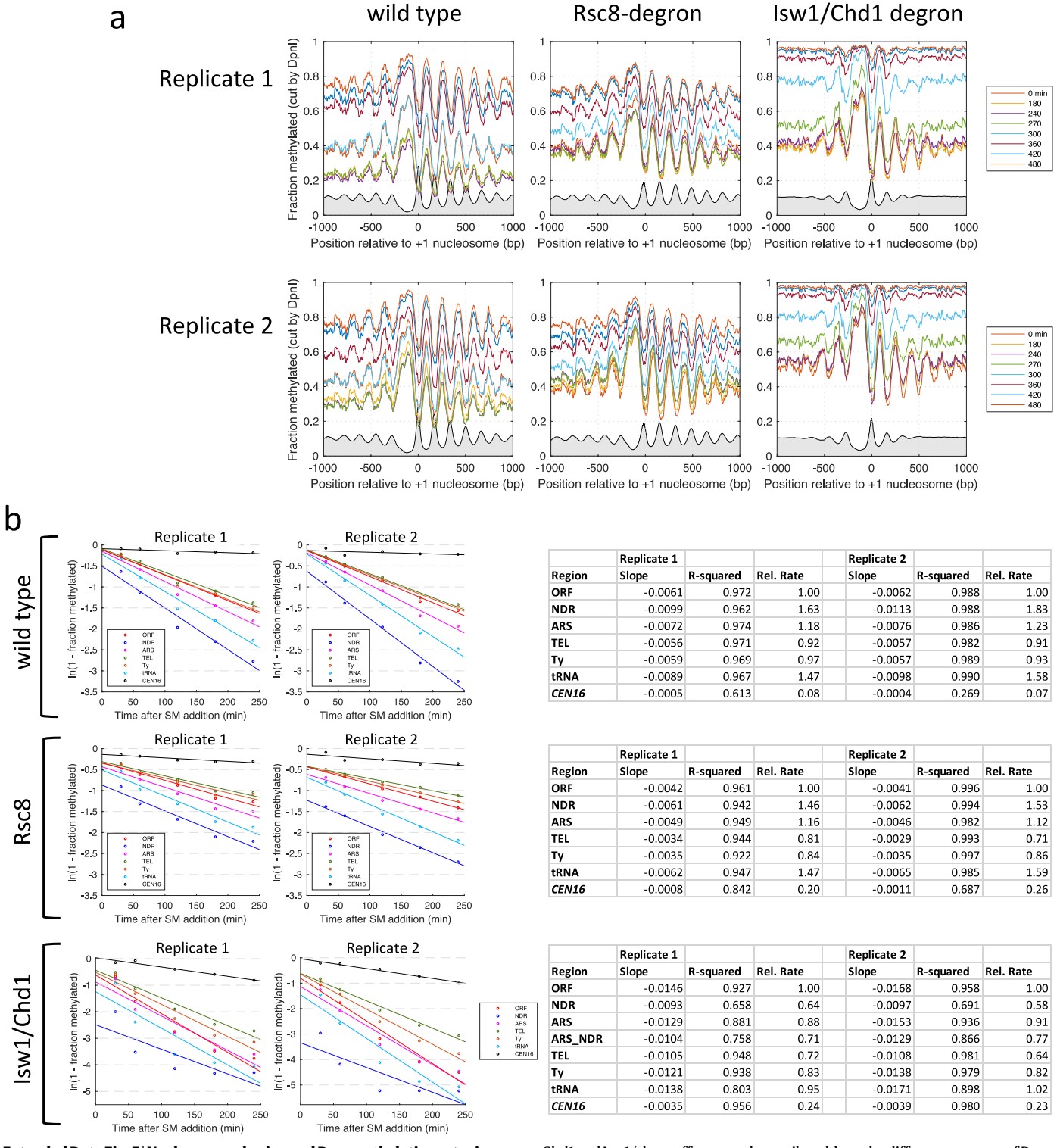

**Extended Data Fig. 7 | Nucleosome phasing and Dam methylation rates in cells depleted of Rsc8 or Isw1 and Chd1.** Cells were grown in auxin for 4 h and then SM was added for a 4 h time course. **a**, Nucleosome phasing as remodelers are depleted. Methylation data obtained at each time point for sites in all genes are plotted relative to the +1 nucleosome dyad (smoothed with a 21-bp window). Grey profiles: nucleosome dyads in the corresponding mutant (MNase-seq data from[26,49] normalised to 0.1). **b**, Genomic regions are generally methylated more slowly in Rsc8-depleted cells than in wild type cells, but faster in cells depleted of

Chd1 and Isw1 (these effects may be attributable to the different amounts of Dam produced in each mutant; see Extended Data Fig. 6a, b). Plots of the natural log of the median unmethylated fraction for GATC sites in each region as a function of time after SM induction. Note the different y-scale used for the Isw1-/Chd1-degron strain. Table: rates relative to ORF sites: slopes and $R^2$ values for different regions for both replicates. The correlation is poor for promoter NDRs in the Isw1/Chd1-degron strain because the median methylation is almost 100% before the time course is complete.

**Extended Data Table 1 | Pearson correlations for 'fcut values' at all GATC sites (Dam experiments) and 'fmeth' values' for all CG sites (M.SssI experiments)**

### Nuclei:

| Dam (units) | Dam (nM) | Nuclei |
|---|---|---|
| 0 | 0 | 0.30 |
| 50 | 0.77 | 0.91 |
| 100 | 1.52 | 0.91 |
| 200 | 2.94 | 0.92 |
| 400 | 5.56 | 0.93 |

### Cells induced with SM (auxin removed):

| Time (min) | WT, Dam | WT, Dam-GFP |
|---|---|---|
| 0 | 0.87 | 0.93 |
| 30 | 0.86 | 0.88 |
| 60 | 0.83 | 0.89 |
| 120 | 0.94 | 0.90 |
| 240 | 0.91 | 0.86 |

### Arrested cells (auxin removed):

| Time (min) | WT, Dam, arrested |
|---|---|
| 0 | 0.89 |
| 180 | 0.91 |
| 210 | 0.93 |
| 240 | 0.94 |
| 300 | 0.93 |
| 420 | 0.93 |

### Remodeler degron mutants (auxin retained):

| Time (min) | WT, auxin | Rsc8-degron | Isw1/Chd1-degrons |
|---|---|---|---|
| 0 | 0.89 | 0.84 | 0.95 |
| 180 | 0.86 | 0.90 | 0.94 |
| 240 | 0.78 | 0.83 | 0.94 |
| 270 | 0.88 | 0.92 | 0.94 |
| 300 | 0.92 | 0.91 | 0.94 |
| 360 | 0.86 | 0.92 | 0.97 |
| 420 | 0.93 | 0.87 | 0.97 |
| 480 | 0.93 | 0.91 | 0.98 |

### Nanopore data: Cells induced with SM (auxin removed):

| Time (min) | WT, M.SssI |
|---|---|
| 0 | 0.82 |
| 30 | 0.85 |
| 60 | 0.89 |
| 120 | 0.96 |
| 240 | 0.94 |
| Neg. control | 0.72 |

Comparison of biological replicate experiments 1 and 2.

# Reporting Summary

## Statistics

For all statistical analyses, confirm that the following items are present in the figure legend, table legend, main text, or Methods section.

| n/a | Confirmed | |
|---|---|---|
| ☒ | ☐ | The exact sample size (*n*) for each experimental group/condition, given as a discrete number and unit of measurement |
| ☐ | ☒ | A statement on whether measurements were taken from distinct samples or whether the same sample was measured repeatedly |
| ☒ | ☐ | The statistical test(s) used AND whether they are one- or two-sided *Only common tests should be described solely by name; describe more complex techniques in the Methods section.* |
| ☒ | ☐ | A description of all covariates tested |
| ☒ | ☐ | A description of any assumptions or corrections, such as tests of normality and adjustment for multiple comparisons |
| ☐ | ☒ | A full description of the statistical parameters including central tendency (e.g. means) or other basic estimates (e.g. regression coefficient) AND variation (e.g. standard deviation) or associated estimates of uncertainty (e.g. confidence intervals) |
| ☒ | ☐ | For null hypothesis testing, the test statistic (e.g. *F*, *t*, *r*) with confidence intervals, effect sizes, degrees of freedom and *P* value noted *Give P values as exact values whenever suitable.* |
| ☒ | ☐ | For Bayesian analysis, information on the choice of priors and Markov chain Monte Carlo settings |
| ☒ | ☐ | For hierarchical and complex designs, identification of the appropriate level for tests and full reporting of outcomes |
| ☐ | ☒ | Estimates of effect sizes (e.g. Cohen's *d*, Pearson's *r*), indicating how they were calculated |

*Our web collection on statistics for biologists contains articles on many of the points above.*

## Software and code

Policy information about availability of computer code

| Data collection | Illumina data were aligned to the sacCer3 version of the yeast genome (available at the Saccharomyces Genome Database) using Bowtie 2. Nanopore data were processed using Nanopolish v.0.14.0, mapped to sacCer3 using Minimap2 v.2.24, and and sorted and indexed with samtools v.1.17. |
|---|---|
| Data analysis | Specific Matlab scripts are publicly available at GitHub: https://github.com/clarkda24/Prajapati_et_al_2023.git |

For manuscripts utilizing custom algorithms or software that are central to the research but not yet described in published literature, software must be made available to editors and reviewers. We strongly encourage code deposition in a community repository (e.g. GitHub). See the Nature Portfolio guidelines for submitting code & software for further information.

## Data

Policy information about availability of data

All manuscripts must include a data availability statement. This statement should provide the following information, where applicable:

- Accession codes, unique identifiers, or web links for publicly available datasets
- A description of any restrictions on data availability
- For clinical datasets or third party data, please ensure that the statement adheres to our policy

All Illumina paired-end and Nanopore sequence data have been deposited in the GEO database with accession number GSE230309 and are publicly available. https://www.ncbi.nlm.nih.gov/geo/query/acc.cgi?acc=GSE230309

# Research involving human participants, their data, or biological material

Policy information about studies with <u>human participants or human data</u>. See also policy information about <u>sex, gender (identity/presentation), and sexual orientation</u> and <u>race, ethnicity and racism</u>.

| | |
|---|---|
| Reporting on sex and gender | No human participants or human data. |
| Reporting on race, ethnicity, or other socially relevant groupings | No human participants or human data. |
| Population characteristics | No human participants or human data. |
| Recruitment | No human participants or human data. |
| Ethics oversight | No human participants or human data. |

Note that full information on the approval of the study protocol must also be provided in the manuscript.

# Field-specific reporting

Please select the one below that is the best fit for your research. If you are not sure, read the appropriate sections before making your selection.

☒ Life sciences ☐ Behavioural & social sciences ☐ Ecological, evolutionary & environmental sciences

For a reference copy of the document with all sections, see <u>nature.com/documents/nr-reporting-summary-flat.pdf</u>

# Life sciences study design

All studies must disclose on these points even when the disclosure is negative.

| | |
|---|---|
| Sample size | Two independent biological replicate experiments were performed for each genomic sample. |
| Data exclusions | Data for GATC sites with neighbouring GATC sites closer than 200 bp on both sides were excluded due to differential loss of short DNA fragments during library preparation. Described in detail in Methods. |
| Replication | Two independent biological replicate experiments were performed for each genomic sample. Parallel analysis of replicate experiments is provided in Extended Data. All replicate experiments were successful. |
| Randomization | Randomisation is not applicable to yeast genomics experiments. |
| Blinding | Blinding is not applicable to yeast genomics experiments |

# Reporting for specific materials, systems and methods

We require information from authors about some types of materials, experimental systems and methods used in many studies. Here, indicate whether each material, system or method listed is relevant to your study. If you are not sure if a list item applies to your research, read the appropriate section before selecting a response.

### Materials & experimental systems

| n/a | Involved in the study |
|---|---|
| ☐ | ☒ Antibodies |
| ☒ | ☐ Eukaryotic cell lines |
| ☒ | ☐ Palaeontology and archaeology |
| ☒ | ☐ Animals and other organisms |
| ☒ | ☐ Clinical data |
| ☒ | ☐ Dual use research of concern |
| ☒ | ☐ Plants |

### Methods

| n/a | Involved in the study |
|---|---|
| ☒ | ☐ ChIP-seq |
| ☒ | ☐ Flow cytometry |
| ☒ | ☐ MRI-based neuroimaging |

## Antibodies

| | |
|---|---|
| Antibodies used | HRP-conjugated anti-HA antibody 3F10 (Roche 12013819001) used at 1:5,000 dilution in 5% skim milk in TBS (20mM Tris-HCl pH 8.0, 0.5 M NaCl) with 0.1% Tween-20 overnight at 4oC. HRP-conjugated anti-tubulin antibody (Abcam ab-185067; lot GR3407868-1) |

diluted 1:20,000 in PBS/0.1% Tween-20 with 0.5% skim milk for 1 h at room temperature with rotation.

Validation

The anti-HA antibody is raised against a haemagglutinin (HA) epitope used for protein tagging. It has been validated by the manufacturer (https://www.sigmaaldrich.com/US/en/product/roche/12013819001#product-documentation). We have observed that this antibody does not give a signal with a yeast strain lacking an HA-tagged protein (as expected). The anti-tubulin antibody was used as a loading control for Western blots. This antibody was validated by the manufacturer for mammalian tubulins (https://www.abcam.com/products/primary-antibodies/hrp-alpha-tubulin-antibody-epr13478b-loading-control-ab185067.html); yeast tubulin is highly conserved and we observed a single band of the expected size (50 kDa).

