## [Peer Review File · Nature Structural & Molecular Biology]

Peer Review Information

Manuscript Title: The yeast genome is globally accessible in living cells

Corresponding author name(s): David Clark

Reviewer Comments & Decisions:

Decision Letter, initial version:
--

Message: 22nd Sep 2023

Dear Dr Clark,

Thank you again for submitting your manuscript "The yeast genome is globally accessible in living cells". I apologize for the delay in responding, which resulted from the difficulty in obtaining suitable referee reports. Nevertheless, we now have comments (below) from the 3 reviewers who evaluated your paper. In light of those reports, we remain interested in your study and would like to see your response to the comments of the referees, in the form of a revised manuscript.

You will see that while Reviewers #1 and #2 appreciate the results and find the conclusions timely and of wide interest, Reviewer #3 finds the work descriptive and confirmatory. While the editorial team has agreed to overrule Reviewer #3's novelty concerns specifically, there are several concerns and suggestions that should be addressed in a revision. With regards to claims made throughout the study about the state of chromatin in living cells, we generally agree with all three Reviewers that there needs to be caution about how the results are interpreted and contextualised, in addition to discussion of alternative interpretations and caveats in the experimental set up, and toning down of some conclusions. We also agree with Reviewers #2 and #3 that expansion of the analysis of the role of different remodellers would strengthen the work and increase its impact and interest to the readership.

Please be sure to address and respond to all concerns of the referees in full in a point-by-point response and highlight all changes in the revised manuscript text file.

We appreciate the requested revisions are extensive. We thus expect to see your revised manuscript within 6 months. If you cannot send it within this time, please let us know. We will be happy to consider your revision as long as nothing similar has been accepted for publication at NSMB or published elsewhere. Should your manuscript be substantially delayed without notifying us in advance and your article is eventually published, the received date would be that of the revised, not the original, version.

Reporting Summary:

When submitting the revised version of your manuscript, please pay close attention to our [href="https://www.nature.com/nature-portfolio/editorial-policies/image-integrity">Digital Image Integrity Guidelines](https://www.nature.com/nature-portfolio/editorial-policies/image-integrity). and to the following points below:

Please note that all key data shown in the main figures as cropped gels or blots should be presented in uncropped form, with molecular weight markers. These data can be aggregated into a single supplementary figure. While these data can be displayed in a relatively informal style, they must refer back to the relevant figures. These data should be submitted with the last revision, prior to acceptance, but you may want to start putting it together at this point.

SOURCE DATA: we urge authors to provide, in tabular form, the data underlying the graphical representations used in figures. This is to further increase transparency in data reporting, as detailed in this editorial (<http://www.nature.com/nsmb/journal/v22/n10/full/nsmb.3110.html>). Spreadsheets can be submitted in excel format. Only one (1) file per figure is permitted; thus, for multi-paneled figures, the source data for each panel should be clearly labeled in the Excel file; alternately the data can be provided as multiple, clearly labeled sheets in an Excel file. When submitting files, the title field should indicate which figure the source data pertains to. We encourage our authors to provide source data at the revision stage, so that they

are part of the peer-review process.

We require deposition of coordinates (and, in the case of crystal structures, structure factors) into the Protein Data Bank with the designation of immediate release upon publication (HPUB). Electron microscopy-derived density maps and coordinate data must be deposited in EMDB and released upon publication. Deposition and immediate release of NMR chemical shift assignments are highly encouraged. Deposition of deep sequencing and microarray data is mandatory, and the datasets must be released prior to or upon publication. To avoid delays in publication, dataset accession numbers must be supplied with the final accepted manuscript and appropriate release dates must be indicated at the galley proof stage. Please find the complete NRG policies on data availability at <http://www.nature.com/authors/policies/availability.html>.

Nature Structural & Molecular Biology is committed to improving transparency in authorship. As part of our efforts in this direction, we are now requesting that all authors identified as 'corresponding author' on published papers create and link their Open Researcher and Contributor Identifier (ORCID) with their account on the Manuscript Tracking System (MTS), prior to acceptance. This applies to primary research papers only. ORCID helps the scientific community achieve unambiguous attribution of all scholarly contributions. You can create and link your ORCID from the home page of the MTS by clicking on 'Modify my Springer Nature account'. For more information please visit please visit www.springernature.com/orcid.

[Redacted]

Sincerely,

Carolina Perdigoto, PhD
Chief Editor
Nature Structural & Molecular Biology
orcid.org/0000-0002-5783-7106

Referee expertise:

Referee #1: chromatin structure and remodelling, genomics, method development, yeast

Referee #2: chromatin, heterochromatin and nuclear organization, yeast

Referee #3: chromatin structure and remodelling, genomics, method development

Reviewers' Comments:

Reviewer #1:

Remarks to the Author:

Summary: The article titled "The yeast genome is globally accessible in living cells" finds that in comparison to isolated nuclei, live yeast cells have a greater overall level of accessible chromatin. First, the authors map accessible chromatin regions in isolated nuclei from budding yeast using the DNA adenine methyltransferase Dam from *E. coli*, which methylates adenines in the GATC motif. From this, the authors show that only a fraction of NDR and ORF sites in nuclei are methylated, even in saturating Dam concentrations, indicating that these sites are protected in a portion of nuclei. In comparison, when Dam is expressed in living cells, the authors find that the fraction of ORFs that are methylated nearly reaches 90% after 240 minutes of Dam induction, indicating chromatin is much more accessible in living cells than isolated nuclei. To look at chromatin accessibility at centromeric regions, the authors replace Dam with the CG methyltransferase M.SssI to gain more methylation sites since only CEN16 contains a single GATC site. The authors note an overall lower level of methylation by M.SssI in ORFs and promoter regions compared to Dam, but centromeric regions at a lower level of methylation than every other site in the genome, indicating they are largely inaccessible.

The authors postulate that the reasoning behind these observed differences in accessibility between nuclei and live cells is due to chromatin remodelers in live cells that lead to fluctuations in chromatin that are absent in nuclei, allowing for global accessibility to be measured *in vivo*. The authors create auxin-inducible degradation systems for three different chromatin remodelers, Rsc8, and Isw1/Chd1, to deplete the living cells of these chromatin remodelers. The degradation of Rsc8 leads to a shift in nucleosome positioning towards the +1 nucleosome. In contrast, degradation of Isw1/Chd1 leads to loss of phasing of nucleosomes around the genic regions and overall increase in accessibility everywhere. The authors' conclusions show that chromatin may not be default repressive in nature but in flux in living cells and not generally repressive. This manuscript is a follow-up to the author's 2019 Genome Research paper, where they found that most of the yeast genome is inaccessible to AluI digestion. The results of the manuscript are expected but highlights the importance of fluctuations in nucleosome positioning in live cells that cannot be captured from isolated nuclei. I recommend this manuscript for publication in Nature Structural and Molecular Biology with the following minor concerns being addressed.

Minor Concerns:

1. The changes in nucleosome positioning through Rsc8 depletion in figures 4C and 4D are slight and may be due to there being a decent amount of protein remaining even in the depletion system. A more convincing argument for changes positioning may come from showing a specific locus where there is a considerable amount of nucleosome shifting towards the +1 position.

2. The resolution for mapping chromatin accessibility is a lot greater for M.SssI because it has over 260,000 sites in the yeast genome compared to Dam which only has ~36,000 sites. M.SssI methylation shows similar trends to Dam methylation but at a lower level,

indicating smaller fluctuations in chromatin accessibility than the Dam methylation data. The authors state the M.SssI methylates slower than Dam and this may be because more of M.SssI sites are occupied in nucleosomes and inaccessible to the methyltransferase.

3. The authors make the statement multiple times that chromatin is globally accessible in living cells, indicating possibly to the reader that many sites are free from nucleosomes when this is not true, but that the nucleosome positioning is fluctuating in living cells, allowing Dam to methylate regions that may be changing in accessibility with transcription. Some of the text should be rewritten to clarify this.

4. In extended figures, 6A and 6B, the wildtype protein level of Dam-3HA differs vastly from the two plots, even though it should be coming from the same wildtype Dam-3HA strain.

5. Extended Data Figure 6 is too large to fit on one page and should be split into two separate figures.

Reviewer #2:

Remarks to the Author:

In this manuscript, the authors report on the genome-wide accessibility of yeast DNA to inducible expression of methyltransferases, Dam and M.Sss.I. Building on their previous work on the accessibility of DNA in nuclei to digestion by restriction enzymes, they show that (1) in isolated nuclei, GATC sites within nucleosomes in ORFs are less accessible than those located in NDRs associated with promoters, (2) in living cells, there is little difference in the end point and rate of GATC methylation between promoter-associated NDRs and ORFs, (3) a GATC within the centromere of chromosome 16 and CG sites within other centromeric nucleosomes are less accessible, and (4) deletion of a subset of chromatin remodelers results in genome-wide shift in nucleosome positions. The experiments in the paper are well designed and executed, the results are of very high technical quality, and support the main conclusions of the paper. Finding #2 above is particularly surprising and will stimulate new thinking and discussion in the field. The paper is in general suitable for publication in NSMB but the authors should address the points below.

1. The authors main conclusion that chromatin is "largely transparent" is primarily based on the comparison of accessibility to Dam for promoter-associated NDRs versus ORFs. They also largely rely on ensemble metagene analysis and with the exception of SCO1 and telomeres do not present other gene-specific accessibility data. One concern is that this analysis may miss the impact of gene-specific pathways on chromatin accessibility. As examples, the authors should compare Dam accessibility (1) between active and repressed genes, such as galactose-induced genes, which under their growth conditions should be repressed, and (2) between SIR-repressed heterochromatic loci and active genes. In the case of gal genes, ideally, they could compare accessibility during growth on glucose versus galactose, but I can see that this may be beyond the scope of the present manuscript. In the case of SIR heterochromatin, they should compare the mating type gene ORFs and promoter regions at HML and HMR with those at the active mat locus (they would need to have HML or HMR delta cells or analyze the surrounding nucleosomes). This analysis would provide insight into how repressive complexes such as the Tup1/Ssn6 and SIR complexes may change chromatin accessibility. In this regard, prior work using the

Dam methylase by Dan Gottschling and the late Amar Klar showed that SIR-repressed regions were less accessible to methylation (PMID: 1570334, PMID: 1737615). These studies performed the above analysis on a smaller gene-specific basis.

2. The authors see global shifts in nucleosome positions in *chd1/isw1*-degron and *rsc8*-degron experiments but no change in accessibility. This seems surprising as one might expect reduced accessibility (at new nucleosome positions). Can the authors explain? Have they tried deletions of other remodelers? Also, differences in Dam expression in wild-type versus mutant cells make these experiments less than ideal.

3. I wonder why the authors observe a greater difference in accessibility between NDRs and ORFs when they use M.SssI compared to Dam. The cen data using M.SssI look great.

Reviewer #3:

Remarks to the Author:

In their submission, "The yeast genome is globally accessible in living cells," Prajapati and colleagues perform *in vivo* measurements of DNA accessibility in dividing and arrested *Saccharomyces cerevisiae* cells using *E. coli* Dam methylase or the CpG methylase M.SssI. They couple DpnI digestion of methylated GATC and Illumina sequencing, or Nanopore sequencing to quantify genomic DNA methylation, observing an increase in bulk chromatin accessibility as measured *in vivo* compared to measurements taken from nuclei. The authors employ a slightly modified version of a previously published technique from their group (qDA-seq), limiting technical novelty of the study. The finding is interesting, though not terribly surprising – even just averaging over the cell cycle, all bases in the yeast genome must be transiently accessible at some point, so the discovery that remodelers contribute to increased accessibility in living, but arrested cells is expected. The manuscript is largely descriptive, with remodeler degron experiments (already performed by others, see PMID30605682) providing confirmatory results.

For these reasons, my enthusiasm for this manuscript is muted. The careful experiments and resulting data will likely be of use to the field, but the manuscript as written suffers from overinterpretation and overemphasis of an obvious conclusion. Moreover, the manuscript fails to account for relevant prior work in the field (see below). In addition to addressing the below comments, I believe a revised manuscript would require a greater breadth of functional experiments (beyond previously-characterized remodeler degrons) and more careful interpretation of resulting data, to be of interest to this or any other journal's broad readership.

Comments

- The fact that nucleosomes are dynamic has been well-established in the literature dating (at least) back to work of Widom, so it is not at all surprising that an *in vivo* labeling approach will measure more accessibility when allowed to label in a living cell, compared to an exogenous approach where (as authors note), concentration of cofactors like ATP likely falls below K_m for respective enzymes. The authors' experiments with remodelers confirms a role for these factors in contributing to observed dynamics (also well-established by prior studies). The experiments are sound, but it is difficult for this reader to identify a non-obvious discovery here.

- One of my biggest issues with the manuscript is the logical jump made by authors from

increased Dam accessibility, to the notion that chromatin is not generally repressive (see last paragraph of discussion and last sentence of abstract). There is abundant data to suggest that chromatin is functionally repressive: first, genetic experiments have directly tied defects in heterochromatin maintenance to retroelement activation and loss of fitness. Second, a multitude of orthogonal assays have confirmed that active regulatory elements across eukarya are more accessible (fewer nucleosomes) compared to inactive regulatory elements, and that the factors that generate this accessibility (as well as the factors that bind regulated, accessible DNA) are genetically essential (see, for instance, the role of mSWI/SNF in regulating NRSF / REST binding in mammalian development). Third (see below), if chromatin were not repressive, then one would expect all yeast TFs to bind all of their high information content motifs across the genome, which does not happen. What the authors have demonstrated here is simply that over time, most bases in the yeast genome are transiently accessible, which seems an obvious conclusion.

- The logical lapse is most apparent on p. 8 line 38. Here, the authors note that Gcn4 can access high affinity sites in gene bodies as well as promoters, but as the authors note in their previously published ChIP-seq study, Gcn4p occupies a subset of motif matches, not all motif matches. By the model presented here by the authors, TFs should nonspecifically bind all motif matches across the genome, independent of context, which is not what is observed in *S. cerevisiae* or in higher eukarya. Thus, my primary concern is that there is still clearly 'regulatory logic' in these cells despite elevated accessibility in vivo, which is not discussed to the detriment of the manuscript.

A revised version of this manuscript should significantly tone down the language used, for example, in the final section of the discussion, the final sentences of the abstract, and the title. Moreover, I would suggest that the authors take greater care to synthesize the considerable amount of work demonstrating the generally repressive nature of chromatin before translating their observation of transient accessibility into a new paradigm for chromatin regulation.

- The authors fail to acknowledge the work of several other groups which are relevant. For instance: PMID31694866 performed methylase-based footprinting to estimate absolute nucleosome occupancy in yeast cells; PMIDs 30017582 and 36001970 employed clever reporter assays to dissect TF binding logic at nucleosome occupied synthetic sequences, with and without remodelers. The manuscript would be much stronger if the authors could integrate their results with prior literature.

Author Rebuttal to Initial comments

Response to Reviewers

We thank the Reviewers for their thoughtful comments and insight, which have led to significant improvements to our manuscript. All changes to the manuscript are in red text.

Reviewer #1:

Remarks to the Author:

Summary: The article titled "The yeast genome is globally accessible in living cells" finds that in comparison to isolated nuclei, live yeast cells have a greater overall level of accessible chromatin. First, the authors map accessible chromatin regions in isolated nuclei from budding yeast using the DNA adenine methyltransferase Dam from E.coli, which methylates adenines in the GATC motif. From this, the authors show that only a fraction of NDR and ORF sites in nuclei are methylated, even in saturating Dam concentrations, indicating that these sites are protected in a portion of nuclei. In comparison, when Dam is expressed in living cells, the authors find that the fraction of ORFs that are methylated nearly reaches 90% after 240 minutes of Dam induction, indicating chromatin is much more accessible in living cells than isolated nuclei. To look at chromatin accessibility at centromeric regions, the authors replace Dam with the CG methyltransferase M.Sss1 to gain more methylation sites since only CEN16 contains a single GATC site. The authors note an overall lower level of methylation by M.Sss1 in ORFs and promoter regions compared to Dam, but centromeric regions at a lower level of methylation than every other site in the genome, indicating they are largely inaccessible.

The authors postulate that the reasoning behind these observed differences in accessibility between nuclei and live cells is due to chromatin remodelers in live cells that lead to fluctuations in chromatin that are absent in nuclei, allowing for global accessibility to be measured in vivo. The authors create auxin-inducible degradation systems for three different chromatin remodelers, Rsc8, and Isw1/Chd1, to deplete the living cells of these chromatin remodelers. The degradation of Rsc8 leads to a shift in nucleosome positioning towards the +1 nucleosome. In contrast, degradation of Isw1/Chd1 leads to loss of phasing of nucleosomes around the genic regions and overall increase in accessibility everywhere. The authors' conclusions show that chromatin may not be default repressive in nature but in flux in living cells and not generally repressive. This manuscript is a follow-up to the author's 2019 Genome Research paper, where they found that most of the yeast genome is inaccessible to AluI digestion. The results of the manuscript are expected but highlights the importance of fluctuations in nucleosome positioning in live cells that cannot be captured from isolated nuclei. I recommend this manuscript for publication in Nature Structural and Molecular Biology with the following minor concerns being addressed.

Minor Concerns:

1. The changes in nucleosome positioning through Rsc8 depletion in figures 4C and 4D are slight and may be due to there being a decent amount of protein remaining even in the depletion system. A more convincing argument for changes positioning may come from showing a specific locus where there is a considerable amount of nucleosome shifting towards the +1 position.

Our MNase-seq studies show that the RSC-dependent shift of the +1 nucleosome towards the promoter is 17 bp (see Fig. 2A in ref. 45 and refs. 47-50). Analysis of the Dam data in Fig. 4d shows that the shift of the +1 nucleosome is 24 +/-1 bp (for 2 replicates). The Dam data are less accurate than the MNase data, because Dam has far fewer sites than MNase. The shift is small, but has been reproduced by many labs, including our own. The low resolution of Dam due to sparse GATC sites means that single gene analysis cannot provide information on the +1 nucleosome shift; combined data from many genes are needed to observe a strong phasing pattern. Analysis of the 1918 genes showing a clear shift of both the +1 and -1

nucleosomes into the promoter that we identified previously (ref. 45: Fig. 4, cluster 1) shows a similar, clear but small RSC-dependent shift (and it was necessary to increase the smoothing window from 21 to 41 nt). We show the data below, but we have not added these plots to the manuscript. We have added a sentence concerning the +1 nucleosome shift on page 7, line 34.

Replicate 1 (cluster 1 genes)

Replicate 2 (cluster 1 genes)

2. The resolution for mapping chromatin accessibility is a lot greater for M.SssI because it has over 260,000 sites in the yeast genome compared to Dam which only has ~36,000 sites. M.SssI methylation shows similar trends to Dam methylation but at a lower level, indicating smaller fluctuations in chromatin accessibility than the Dam methylation data. The authors state the M.SssI methylates slower than Dam and this may be because more of M.SssI sites are occupied in nucleosomes and inaccessible to the methyltransferase.

The methylation rate depends on the amount of substrate (methylation sites), the enzyme concentration and the intrinsic properties of the enzyme. We have amended the text on page 5, line 40 stating: "The different methylation rates exhibited by Dam and M.SssI could be due to differences in enzyme concentration, enzyme turnover number, and/or the number of potential methylation sites." We also note that M.SssI is starting at a much lower baseline than Dam (the latter has a significantly higher background methylation in the absence of SM); please see our reply to Referee 2 point 3, and the additional text on page 5, line 41.

3. The authors make the statement multiple times that chromatin is globally accessible in living cells, indicating possibly to the reader that many sites are free from nucleosomes when this is not true, but that the nucleosome positioning is fluctuating in living cells, allowing Dam to methylate regions that may be changing in accessibility with transcription. Some of the text should be rewritten to clarify this. We certainly do not want to give the impression that many GATC sites are nucleosome-free, but we are not sure where the text is unclear. Please give specific examples and we will clarify the text. We hope that the reader will understand the potential sources of genome accessibility by referring to Fig. 2i.

4. In extended figures, 6A and 6B, the wildtype protein level of Dam-3HA differs vastly from the two plots, even though it should be coming from the same wildtype Dam-3HA strain. Actually, this is not the case. The same set of wild type samples was loaded in the Rsc8 and Isw1/Chd1 blots and so the wild type data should have been normalised to 1. We apologise for this error. We have corrected the figure, by setting the maximum wild type signal to 1 in both plots.

5. Extended Data Figure 6 is too large to fit on one page and should be split into two separate figures. We have split Extended Data Fig. 6 into two separate figures.

Reviewer #2:

Remarks to the Author:

In this manuscript, the authors report on the genome-wide accessibility of yeast DNA to inducible expression of methyltransferases, Dam and M.Sss.I. Building on their previous work on the accessibility of DNA in nuclei to digestion by restriction enzymes, they show that (1) in isolated nuclei, GATC sites within nucleosomes in ORFs are less accessible than those located in NDRs associated with promoters, (2) in living cells, there is little difference in the end point and rate of GATC methylation between promoter-associated NDRs and ORFs, (3) a GATC within the centromere of chromosome 16 and CG sites within other centomeric nucleosomes are less accessible, and (4) deletion of a subset of chromatin remodelers results in genome-wide shift in nucleosome positions. The experiments in the paper are well designed and executed, the results are of very high technical quality, and support the main conclusions of the paper. Finding #2 above is particularly surprising and will stimulate new thinking and discussion in the field. The paper is in general suitable for publication in NSMB but the authors should address the points below.

1. The authors main conclusion that chromatin is “largely transparent” is primarily based on the comparison of accessibility to Dam for promoter-associated NDRs versus ORFs. They also largely rely on ensemble metagene analysis and with the exception of SCO1 and telomeres do not present other gene-specific accessibility data. One concern is that this analysis may miss the impact of gene-specific pathways on chromatin accessibility. As examples, the authors should compare Dam accessibility (1) between active and repressed genes, such as galactose-induced genes, which under their growth conditions should be repressed, and (2) between SIR-repressed heterochromatic loci and active genes. In the case of gal genes, ideally, they could compare accessibility during growth on glucose versus galactose, but I can see that this may be beyond the scope of the present manuscript. In the case of SIR heterochromatin, they should compare the mating type gene ORFs and promoter regions at HML and HMR with those at the active mat locus (they would need to have HML or HMR delta cells or analyze the surrounding nucleosomes). This analysis would provide insight into how repressive complexes such as the Tup1/Ssn6 and SIR complexes may change chromatin accessibility. In this regard, prior work using the Dam methylase by Dan Gottschling and the late Amar Klar showed that SIR-repressed regions were less accessible to methylation (PMID: 1570334, PMID: 1737615). These studies performed the above analysis on a smaller gene-specific basis.

Our main conclusion that chromatin is largely transparent rests primarily on the observation that virtually all GATC sites are methylated within a relatively short time frame, unlike in nuclei. We point out that the ORF accessibility plot (Fig. 2e) shows the range of data that includes 90% of all GATC sites located within ORFs; this range is quite narrow, indicating that nearly all genes are accessible in living cells. However, we agree that we might be missing some gene-specific effects.

In the case of the *GAL* genes, there are 21 GATC sites located within the *GAL1/10/7* ORFs. These genes should be repressed under our growth conditions (glucose medium). However, analysis of these sites relative to all ORF sites shows little or no difference in median methylation rate in wild type cells, although there is quite a wide range in site methylation rate. These data suggest that the repressed *GAL* genes are accessible.

Dam methylation in the *GAL1*, *GAL7* and *GAL10* ORFs. a, Median methylation and range for the 21 GATC sites in these ORFs. b, Median methylation of *GAL* ORFs compared with that of all ORFs.

We decided not to repeat the experiment with cells grown in galactose, because that would require careful comparison of the amounts of Dam produced in glucose and galactose, and because there are only 3 affected genes. Instead, we compared the methylation rates for SM/Gcn4-activated genes with those for all other genes in the same wild type data sets (internally controlled for Dam concentration). We find that the 2496 GATC sites within the ORFs of 512 Gcn4-activated genes (ref. 27) are methylated marginally, but reproducibly, faster than other genes. We also compared a subset of 235 GATC sites within 49 Gcn4-activated ORFs with highly disrupted chromatin structure (including some H2A-H2B dimer loss) correlated with high Pol II levels (refs. 28, 29). These genes are methylated faster than the complete set of Gcn4 target genes, indicating that very highly transcribed genes are more open. We have added the plots as Extended Data Fig. 3a and explanatory text on page 4, line 26.

The Reviewer suggested that we analyse SIR-repressed heterochromatin. As the Reviewer points out, this is tricky because the active *MAT* locus is identical in sequence to one of the two silenced loci and therefore the two loci cannot be distinguished using the Dam short read data. An additional problem is the dearth of GATC sites in these loci. However, the long reads obtained in our M.SssI Nanopore data include flanking sequences, allowing us to separate data for the active *MAT* locus and the silenced locus, and with the higher resolution associated with CpG methylation. Our strain is *MAT α* whereas the reference genome is *MAT α* . Consequently, we enlisted the help of another postdoc in the lab, Zhuwei Xu, who created a new *MAT α* reference genome, aligned our Nanopore data to it, and analysed the active *MAT α* locus and both silenced loci (*HML α* and *HMR α*). He found that the silenced loci are methylated much more slowly than the active *MAT α* locus. We have added the plot for replicate 1 as Fig. 3f and the plots for both replicates as Extended Data Fig. 5b. We adjusted the Abstract and the Methods (page 20, line 26), and added explanatory text (page 6, line 18). We have included Zhuwei as an author to reflect his important contribution. We have also cited the two papers mentioned by the Reviewer (refs. 36, 37; they compare relative Dam methylation at single loci using one methylation time point). We thank the Reviewer for suggesting analysis of SIR-repressed heterochromatin.

2. The authors see global shifts in nucleosome positions in *chd1/isw1*-degron and *rsc8*-degron experiments but no change in accessibility. This seems surprising as one might expect reduced accessibility (at new nucleosome positions). Can the authors explain? Have they tried deletions of other remodelers? Also, differences in Dam expression in wild-type versus mutant cells make these experiments less than ideal.

When we began the remodeler experiments, we hypothesised that loss of the remodelers responsible would prevent nucleosome flux, resulting in a limit level of methylation well short of complete methylation, similar to the result for nuclei. We did not observe such a result when either RSC or ISW1/CHD1 were depleted. Consequently, we propose that multiple remodelers contribute to the flux. Determination of remodeler contributions to nucleosome flux is complicated for two reasons: (1) As the Reviewer points out, there are differences in Dam expression: *Rsc8*-depleted cells produce less Dam than wild type, whereas *Isw1/Chd1*-depleted cells produce more Dam than wild type (Extended Data Fig. 6a/b). Thus, the methylation rate plots for different strains cannot be compared directly without normalising for Dam concentration, which changes during the time course (unlike in a standard in vitro enzyme kinetics experiment), and in a strain-dependent manner. (2) While most of the remodeler is depleted when Dam is induced, residual remodeler may be sufficient to maintain flux. Ideally, all of the remodeler would be depleted before Dam is induced, but the loss of the remodeler affects Dam production. Consequently, we focused on RSC and ISW1/CHD1 because MNase-seq experiments show global changes in chromatin organisation in nuclei from depleted or null cells. We reasoned that we should be able to observe the chromatin change in living cells as the remodeler is depleted, and this was the case (Fig. 4). The effects of other remodelers on global chromatin structure, as measured by MNase-

seq, are more subtle (INO80), or absent (SWI/SNF). Therefore, we did not expect to obtain evidence for nucleosome flux by depleting these remodelers. We have preliminary data confirming that loss of Ino80 or Snf2 also alters Dam levels relative to wild type, without significant changes in global nucleosome phasing (as expected from MNase-seq data).

3. I wonder why the authors observe a greater difference in accessibility between NDRs and ORFs when they use M.SssI compared to Dam. The cen data using M.SssI look great.

This is an interesting question. We think that the explanation lies in the difference in background methylation (before induction). In the absence of SM induction, Dam methylation is quite high, especially at NDRs (Fig. 2e), whereas M.SssI methylation is very low (Fig. 3c). We propose that, at any moment in time, the NDR belonging to each gene is virtually protein-free in a fraction of cells and occupied by a non-histone complex in the remaining cells (ref. 22). If the NDR is protein-free, it is methylated rapidly even at very low Dam levels (M.SssI levels are too low) whereas, if the NDR is occupied, methylation is slower, occurring only when the occupying complex dissociates. If correct, then the Dam rate at NDRs is determined by the slower NDR complex turnover rate, because the protein-free NDRs have already been methylated. In contrast, the M.SssI rate is composed of two different rates, for protein-free NDRs and complexed NDRs, with an average that is faster than the Dam rate. We have amended the text on page 5, line 41.

Reviewer #3:

Remarks to the Author:

In their submission, “The yeast genome is globally accessible in living cells,” Prajapati and colleagues perform in vivo measurements of DNA accessibility in dividing and arrested *Saccharomyces cerevisiae* cells using *E. coli* Dam methylase or the CpG methylase M.SssI. They couple DpnI digestion of methylated GATC and Illumina sequencing, or Nanopore sequencing to quantify genomic DNA methylation, observing an increase in bulk chromatin accessibility as measured in vivo compared to measurements taken from nuclei. The authors employ a slightly modified version of a previously published technique from their group (qDA-seq), limiting technical novelty of the study. The finding is interesting, though not terribly surprising – even just averaging over the cell cycle, all bases in the yeast genome must be transiently accessible at some point, so the discovery that remodelers contribute to increased accessibility in living, but arrested cells is expected. The manuscript is largely descriptive, with remodeler degnon experiments (already performed by others, see PMID30605682) providing confirmatory results.

For these reasons, my enthusiasm for this manuscript is muted. The careful experiments and resulting data will likely be of use to the field, but the manuscript as written suffers from overinterpretation and overemphasis of an obvious conclusion. Moreover, the manuscript fails to account for relevant prior work in the field (see below). In addition to addressing the below comments, I believe a revised manuscript would require a greater breadth of functional experiments (beyond previously-characterized remodeler degnons) and more careful interpretation of resulting data, to be of interest to this or any other journal’s broad readership.

1. We are pleased that the Reviewer finds our study interesting. However, we are unsure why it is not surprising or why the Reviewer considers the result to be obvious, given that the generally accepted model of gene regulation depends on limited DNA accessibility in chromatin. We agree that all bases in the genome are likely to be transiently accessible during DNA replication, but this is not necessarily true of arrested cells, which is why we did that experiment. In any case, our paper describes the first quantitative measurements of genome accessibility in living cells. We think most of our colleagues would

have supposed that accessibility in cells would not differ much from that in nuclei (as we did). If so, the result is surprising - and requires some reappraisal of current models.

We have now cited the excellent study by Klein-Brill et al. (ref 50), who used remodeler degron strains to perform time courses of depletion and recovery of remodeler activity, followed by MNase-seq, to show that global RSC-mediated nucleosome shifts are reversible. As concluded in earlier MNase-seq studies involving RSC subunit depletion (refs. 45, 47-50), it can be inferred that global nucleosome shifts occurred within the cells during the time course. However, this type of study does not distinguish between two scenarios: (1) a shift from one set of static nucleosome positions to another set of static positions, or (2), a net shift in the average positions of nucleosomes that are in continuous flux (as we have shown). More importantly, this type of experiment does not provide information on nucleosome dynamics in wild type cells - if a mock depletion experiment had been done with wild type cells, no changes in chromatin organisation would be observed before, during or after mock depletion, since MNase-seq measures the static situation in nuclei.

Comments

- The fact that nucleosomes are dynamic has been well-established in the literature dating (at least) back to work of Widom, so it is not at all surprising that an *in vivo* labeling approach will measure more accessibility when allowed to label in a living cell, compared to an exogenous approach where (as authors note), concentration of cofactors like ATP likely falls below K_m for respective enzymes. The authors' experiments with remodelers confirms a role for these factors in contributing to observed dynamics (also well-established by prior studies). The experiments are sound, but it is difficult for this reader to identify a non-obvious discovery here.

2. It is true that nucleosome dynamics have been studied for many years. Jon Widom's work is certainly illuminating, but his studies and those that followed show that while the nucleosomal DNA ends (10-20 bp) transiently dissociate and re-associate from the histone octamer on a fast time scale, the central core of the nucleosome is essentially stable and inaccessible *in vitro* (extreme concentrations of restriction enzymes are required to cut a site near the dyad). We have added a comment on page 7, line 45. Many more recent studies show that remodelers move nucleosomes *in vitro*, although there is generally no specificity for DNA sequence (this is of course a standard assay for remodelling activity). But *in vivo*, most studies infer that remodelers are recruited to specific promoters where they create and maintain a nucleosome-depleted region (NDR). They do not postulate the global accessibility that we observe. Using the methylase approach, we have detected global movements of nucleosomes in wild type cells and we have also provided kinetic information, resulting from our quantitative approach. There is no other study like ours.

- One of my biggest issues with the manuscript is the logical jump made by authors from increased Dam accessibility, to the notion that chromatin is not generally repressive (see last paragraph of discussion and last sentence of abstract). There is abundant data to suggest that chromatin is functionally repressive: first, genetic experiments have directly tied defects in heterochromatin maintenance to retroelement activation and loss of fitness. Second, a multitude of orthogonal assays have confirmed that active regulatory elements across eukarya are more accessible (fewer nucleosomes) compared to inactive regulatory elements, and that the factors that generate this accessibility (as well as the factors that bind regulated, accessible DNA) are genetically essential (see, for instance, the role of mSWI/SNF in regulating NRSF / REST binding in mammalian development). Third (see below), if chromatin were not repressive, then one would expect all yeast TFs to bind all of their high information content motifs across the genome, which does not happen. What the authors have demonstrated here is simply that over time, most bases in the yeast genome are transiently accessible, which seems an obvious conclusion.

3. We think the issue here is that we failed to point out that yeast chromatin is almost all euchromatin, whereas the examples provided by the Reviewer apply mostly to the heterochromatin of higher organisms. Furthermore, in response to Reviewer 2, we have added an analysis of the silenced loci, showing that silenced loci are almost as resistant to methylation as centromeres (Fig. 3f). These silenced genes may be more similar to the heterochromatic genes of higher organisms. We have added the following sentence on page 9, line 35: "We note that yeast chromatin is essentially equivalent to euchromatin in the cells of higher organisms (with the exception of the silenced loci). In higher organisms, repressed genes generally reside in facultative heterochromatin, which may resemble the yeast silenced loci, and so may well have different flux properties from euchromatin."

To discuss the individual points raised: (1) There is very little heterochromatin in budding yeast. We have shown in new Fig. 3f that SIR-dependent heterochromatin is methylated much more slowly than other genomic regions. However, SIR-dependent heterochromatin is formed only on a tiny proportion of yeast genes (< 1%). The cells of higher organisms have far more extensive heterochromatin. Therefore, we believe it is fair to state that the yeast genome is generally accessible. (2) We agree that, in *nuclei*, active regulatory elements are more accessible than inactive elements, due to decreased nucleosome occupancy and static chromatin. However, in living yeast cells, we find that the accessibilities of ORF DNA and promoter NDRs are quite similar, because both nucleosomes and non-histone complexes are moving, or being removed and replaced, at a high rate. As mentioned above, the heterochromatin of higher organisms may well differ from yeast euchromatin. (3) We agree that "if chromatin were not repressive, then one would expect all yeast TFs to bind all of their high information content motifs across the genome", but we do not agree that "it does not happen". In a previous paper (ref. 24), we have shown that it does happen, at least for the Gcn4 transcription factor (please see below).

Finally, we argue that it is obvious that, if sequence-specific DNA methylases can penetrate the chromatin and mark the DNA within one cell cycle, then transcription factors can also access their binding sites and regulate genes. Hence the conclusion that the genome is globally accessible and that DNA packaging is not generally repressive, unless heterochromatin factors are present, such as the Sir proteins.

- The logical lapse is most apparent on p. 8 line 38. Here, the authors note that Gcn4 can access high affinity sites in gene bodies as well as promoters, but as the authors note in their previously published ChIP-seq study, Gcn4p occupies a subset of motif matches, not all motif matches. By the model presented here by the authors, TFs should nonspecifically bind all motif matches across the genome, independent of context, which is not what is observed in *S. cerevisiae* or in higher eukarya. Thus, my primary concern is that there is still clearly 'regulatory logic' in these cells despite elevated accessibility *in vivo*, which is not discussed to the detriment of the manuscript.

4. If chromatin is unimportant *in vivo* in determining binding by a transcription factor such as Gcn4, we would expect that Gcn4 would bind its sites according to the relative affinities of its various motif variants. The relative occupancies of these sites will be governed by the Gcn4 concentration. Unless the Gcn4 concentration is very high, we would predict that high-affinity sites would give much stronger ChIP signals than low-affinity sites. This is what we observed in our paper (ref. 24), which follows on from our earlier ChIP-seq study for Gcn4 mentioned by the Reviewer (ref. 64). We identified an extended motif corresponding to a subset of high-affinity Gcn4 sites and found that Gcn4 occupies *all* of these high-affinity sites in the yeast genome, irrespective of whether they are in promoters or in ORFs, and averaging only about 2-fold lower occupancy at ORF sites relative to NDR sites. The low-affinity sites are almost all low occupancy, as would be expected given their low affinity. Please see Fig. 6 in ref. 24. We propose that sequence-specific transcription factors are the primary drivers of gene regulation in yeast; they bind when a site is exposed during flux, and activate or repress genes according to their

concentrations, motif affinity and gene-specific feedback systems. We also note that pioneer factors would not have to wait for site exposure. We have added comments on page 9, line 29. We thank the Reviewer for highlighting this issue.

A revised version of this manuscript should significantly tone down the language used, for example, in the final section of the discussion, the final sentences of the abstract, and the title. Moreover, I would suggest that the authors take greater care to synthesize the considerable amount of work demonstrating the generally repressive nature of chromatin before translating their observation of transient accessibility into a new paradigm for chromatin regulation.

5. We believe that our title represents a fair conclusion from our data, and it does refer to yeast (see above). The Reviewer did not dispute our conclusion, or question our data, instead stating that it is obvious. In the Abstract, we are "suggesting" that DNA packaging is not generally repressive - we don't think this statement is excessive for a final sentence. We believe that our findings suggest that a new paradigm for chromatin regulation may be necessary, although we are not proposing one here. We are adding an important new fact to the debate. After careful consideration of the Reviewer's arguments, we realised that we should have mentioned that yeast chromatin is essentially equivalent to euchromatin in the cells of higher organisms (with the exception of the silenced loci). We have amended the Abstract (by inserting the words "in yeast") and the end of the main text (by adding the final sentence quoted above).

- The authors fail to acknowledge the work of several other groups which are relevant. For instance: PMID31694866 performed methylase-based footprinting to estimate absolute nucleosome occupancy in yeast cells; PMIDs 30017582 and 36001970 employed clever reporter assays to dissect TF binding logic at nucleosome occupied synthetic sequences, with and without remodelers. The manuscript would be much stronger if the authors could integrate their results with prior literature.

6. We should have cited Oberbeckmann et al. (now cited as ref 21; comment inserted on page 3, line 25). They treated yeast nuclei (not living cells) with other DNA methylases as well as restriction enzymes; their nucleosome occupancy measurements are consistent with our data for yeast nuclei (ref. 20 and this study). The very nice papers from Lu Bai's lab concerning transcription factor binding in yeast seem to be only tangentially relevant to our study. In the second paper, which is essentially a footprinting study, a DNA methylase (M.CviPI) was expressed in vivo; one methylation time point was used to investigate pioneer transcription factor binding at a single artificial locus. Although we do not cite these papers because an extensive digression would be required, we have added some comments concerning transcription factors and gene regulation on page 9 line 29 (also mentioned above).

Decision Letter, first revision:

Message: Our ref: NSMB-A47763A

16th Feb 2024

Dear Dr. Clark,

Thank you for submitting your revised manuscript "The yeast genome is globally accessible in living cells" (NSMB-A47763A). It has now been seen by the original referees and their comments are below. The reviewers find that the paper has improved in revision, and therefore we'll be happy in principle to publish it in Nature Structural & Molecular Biology, pending minor revisions to satisfy the referees' final requests and to comply with our editorial and formatting guidelines.

To facilitate our work at this stage, it is important that we have a copy of the main text as a word file. If you could please send along a word version of this file as soon as possible, we would greatly appreciate it; please make sure to copy the NSMB account (cc'ed above).

Sincerely,

Carolina Perdigoto, PhD
Chief Editor
Nature Structural & Molecular Biology
orcid.org/0000-0002-5783-7106

Reviewer #1 (Remarks to the Author):

The authors have addressed my concerns and I support publication.

Reviewer #3 (Remarks to the Author):

Given the suggestions of Reviewers 1 and 2 leading to the authors' additional experimentation, I do not have specific experimental / analysis suggestions. In fact, I believe a version of this text that is sufficiently-revised to tone down claims would be of great interest to the community and this journal's readership! However, I do not believe the manuscript as written appropriately serves the chromatin field.

Specifically, the manuscript still overstates claims and interpretations from this "hyperaccessible" result, especially given the data presented in response to Reviewer 2, which demonstrates that genome accessibility *is* regulated in the context of SIR-mediated silencing (even in a genome that is basically all euchromatin). Virtually all of my issues with the manuscript lie in the final paragraph of the discussion (beginning on p. 9 l. 13), entitled "Implications for gene regulation." I enumerate these issues below:

Comments:

1.) The discussion mentions enhancers (p.9 l.16), but as the authors mention in their last rebuttal, this article focuses on *yeast chromatin* which does not have enhancers, and which is mostly euchromatin. This sentence should be deleted, lest readers confuse the highly-specific findings of this manuscript for more general principles of gene regulation (for instance, in metazoa).

2.) p.9 l.21 "This model posits...do not spread to gene bodies" is factually incorrect and does not accurately describe a model held in the field re: ATP-dependent chromatin remodeling. Remodelers do act in gene bodies, in concert with e.g. transcribing RNA Polymerase II (there are many citations, but PMID22726434 classically showed this genome-wide).

3.) The authors continue to overstate claims with words like 'transparent.' To this reader, the model best supported by their data is: ATP-dependent chromatin dynamics exist *in vivo*, and these dynamics mean that per-base accessibility over short periods of time for much of the yeast genome (except silenced regions) is at some point 1 / 100% / open. This is *interesting*, but is by no means synonymous with 'transparency.' Nucleosomes are critical and deeply-conserved receptors for a variety of essential processes, and they serve to repress by (among other things) competing with other factors for access to DNA. Nothing in the authors' manuscript runs counter to this; rather, this manuscript serves as a nice illustration of how dynamic nucleosomes can be *in vivo*. I fear that the authors' insistence on using simple terms to describe their results will only add confusion to the field; I would suggest that the authors edit this section and replace terms like 'transparent' with more honest descriptions of their results.

4.) p.9 l.27 "On the other hand..." What evidence have the authors provided that nucleosome flux may dislodge transcription factors? ATP-dependent chromatin remodelers have well-understood docking sites on the nucleosome itself. Are the authors suggesting that remodelers may also evict TFs? If so, this is a confusing and somewhat bold statement to make given the lack of any data supporting this notion.

5.) p.9 l.33 "In conclusion..." I am confused by the authors insistence that 'virtually all' yeast chromatin is in this continuous state of flux; have the authors not shown in this revision that silent mating type loci and centromeres are quantitatively different with regards to their *in vivo* accessibility? This should be reconciled in a revised discussion, because it demonstrates that chromatin accessibility is in fact regulated, even in the context of the highly-compact and mostly euchromatic yeast genome.

Author Rebuttal, first revision:

NSMB-A47763A

We thank the Reviewers for their detailed and thoughtful reviews, which have significantly improved our manuscript.

Our replies to Reviewer 3 are in blue text below.

Reviewer #1 (Remarks to the Author):

The authors have addressed my concerns and I support publication.

Reviewer #3 (Remarks to the Author):

Given the suggestions of Reviewers 1 and 2 leading to the authors' additional experimentation, I do not have specific experimental / analysis suggestions. In fact, I believe a version of this text that is sufficiently-revised to tone down claims would be of great interest to the community and this journal's readership! However, I do not believe the manuscript as written appropriately serves the chromatin field.

Specifically, the manuscript still overstates claims and interpretations from this "hyperaccessible" result, especially given the data presented in response to Reviewer 2, which demonstrates that genome accessibility *is* regulated in the context of SIR-mediated silencing (even in a genome that is basically all euchromatin). Virtually all of my issues with the manuscript lie in the final paragraph of the discussion (beginning on p. 9 l. 13), entitled "Implications for gene regulation." I enumerate these issues below:

As requested, we have made several changes to the final paragraph, detailed below.

Comments:

1.) The discussion mentions enhancers (p.9 l.16), but as the authors mention in their last rebuttal, this article focuses on *yeast chromatin* which does not have enhancers, and which is mostly euchromatin. This sentence should be deleted, lest readers confuse the highly-specific findings of this manuscript for more general principles of gene regulation (for instance, in metazoa).

This is a fair comment. We stated: "It is envisaged that regulatory elements, such as promoters and enhancers, are blocked by nucleosomes and perhaps by the formation of higher-order chromatin structures and heterochromatin, preventing transcription factors from binding to their cognate sites and activating transcription." We have re-written this sentence to remove the reference to enhancers: "It is envisaged that promoters are blocked by nucleosomes, preventing transcription factors from binding to their cognate sites and activating transcription."

2.) p.9 l.21 "This model posits...do not spread to gene bodies" is factually incorrect and does not

accurately describe a model held in the field re: ATP-dependent chromatin remodeling. Remodelers do act in gene bodies, in concert with e.g. transcribing RNA Polymerase II (there are many citations, but PMID22726434 classically showed this genome-wide).

We agree with the Reviewer that this sentence is ambiguous. We meant that, in multiple reviews, models describing the chromatin structure of the typical gene before and after activation propose major changes in chromatin structure only at promoters. Accordingly, we have removed the phrase: "which do not spread to gene bodies".

3.) The authors continue to overstate claims with words like 'transparent.' To this reader, the model best supported by their data is: ATP-dependent chromatin dynamics exist in vivo, and these dynamics mean that per-base accessibility over short periods of time for much of the yeast genome (except silenced regions) is at some point 1 / 100% / open. This is *interesting*, but is by no means synonymous with 'transparency.' Nucleosomes are critical and deeply-conserved receptors for a variety of essential processes, and they serve to repress by (among other things) competing with other factors for access to DNA. Nothing in the authors' manuscript runs counter to this; rather, this manuscript serves as a nice illustration of how dynamic nucleosomes can be in vivo. I fear that the authors' insistence on using simple terms to describe their results will only add confusion to the field; I would suggest that the authors edit this section and replace terms like 'transparent' with more honest descriptions of their results.

We believe that our description of our results is honest. We use the word "transparent" to communicate what we believe our observations clearly show. We think the model proposed by the Reviewer is consistent with transparency: the genome is open some of the time, enough to allow a sequence-specific DNA methylase like Dam to access the DNA during a single cell cycle. We would argue that if Dam can gain access, so can a sequence-specific transcription factor. The nucleosome block model is only viable if nucleosomes block access to the DNA at all times. By using the word "transparent", we are conveying the implication of global accessibility.

4.) p.9 l.27 "On the other hand..." What evidence have the authors provided that nucleosome flux may dislodge transcription factors? ATP-dependent chromatin remodelers have well-understood docking sites on the nucleosome itself. Are the authors suggesting that remodelers may also evict TFs? If so, this is a confusing and somewhat bold statement to make given the lack of any data supporting this notion.

We stated: "On the other hand, the same nucleosome flux that creates transient access might also dislodge bound transcription factors from their sites, reducing their residence times, perhaps to the point where their binding is non-functional." This sentence is part of our discussion of potential consequences of our chromatin flux model. It is clear that we are not claiming to have shown this. However, we decided to delete this sentence.

5.) p.9 l.33 "In conclusion..." I am confused by the authors insistence that 'virtually all' yeast chromatin is in this continuous state of flux; have the authors not shown in this revision that silent mating type loci and centromeres are quantitatively different with regards to their in vivo accessibility? This should be reconciled in a revised discussion, because it demonstrates that

chromatin accessibility is in fact regulated, even in the context of the highly-compact and mostly euchromatic yeast genome.

It is accurate and honest to use the words "virtually all", since the inaccessible regions represent < 0.1% of the yeast genome. The yeast genome is about 12.1 Mb. There is a single centromeric nucleosome on each of the 16 chromosomes = 16x 150 bp = 2,400 bp. There are two silenced mating type loci: HMRA (2647 bp) and HMLalpha (3704 bp). Total inaccessible chromatin = 8.8 kb, or < 0.1% of the genome. We have added an explanatory sentence on page 9 line 32: "The exceptions are the silenced loci (HMRA and HMLalpha) and the sixteen centromeric nucleosomes (one on each chromosome), which together account for < 0.1% of the yeast genome."

Final Decision Letter:

Message: 17th Apr 2024

Dear Dr. Clark,

We are now happy to accept your revised paper "The yeast genome is globally accessible in living cells" for publication as an Article in Nature Structural & Molecular Biology.

To assist our authors in disseminating their research to the broader community, our SharedIt initiative provides all co-authors with the ability to generate a unique shareable

link that will allow anyone (with or without a subscription) to read the published article. Recipients of the link with a subscription will also be able to download and print the PDF.

Your paper will be published online soon after we receive proof corrections and will appear in print in the next available issue. You can find out your date of online publication by contacting the production team shortly after sending your proof corrections.

You may wish to make your media relations office aware of your accepted publication, in case they consider it appropriate to organize some internal or external publicity. Once your paper has been scheduled you will receive an email confirming the publication details. This is normally 3-4 working days in advance of publication. If you need additional notice of the date and time of publication, please let the production team know when you receive the proof of your article to ensure there is sufficient time to coordinate. Further information on our embargo policies can be found here:
<http://www.nature.com/authors/policies/embargo.html>

Please note that *Nature Structural & Molecular Biology* is a Transformative Journal (TJ). Authors may publish their research with us through the traditional subscription access route or make their paper immediately open access through payment of an article-

processing charge (APC). Authors will not be required to make a final decision about access to their article until it has been accepted. Find out more about Transformative Journals

Sincerely,

Dimitris Typas
Associate Editor
Nature Structural & Molecular Biology
ORCID: 0000-0002-8737-1319